



# Basic controller tuning for large offshore wind turbines

Karl O. Merz

SINTEF Energy Research, Sem Sælandsvei 11, 7034 Trondheim, Norway

*Correspondence to*: Karl O. Merz (karl.merz@sintef.no)

**Abstract.** When a wind turbine operates above the rated windspeed, the blade pitch may be governed by a basic single-input single-output PI controller, with the shaft speed as input. The performance of the wind turbine depends upon the tuning of the gains and filters of this controller. Rules-of-thumb, based upon pole placement, with a rigid model of the rotor, are inadequate for tuning the controller of large, flexible, offshore wind turbines. It is shown that the appropriate controller tuning is highly dependent upon the characteristics of the aeroelastic model: no single reference controller can be defined for

use with all models. As an example, the ubiquitous NREL 5 MW wind turbine controller is unstable, when paired with a fully-flexible aeroelastic model. A methodical search is conducted, in order to find models with a minimum number of degrees-of-freedom, which can be used to tune the controller for a fully-flexible aeroelastic model; this can be accomplished with a model containing 16-20 states. Transient aerodynamic effects, representing rotor-average properties, account for 5 of these states. A simple method is proposed to reduce the full transient aerodynamic model, and the associated turbulent wind

spectra, to the rotor-average. Ocean waves are also an important source of loading; it is recommended that the shaft speed signal be filtered such that wave-driven tower side-to-side vibrations do not appear in the PI controller output. An updated tuning for the NREL 5 MW controller is developed using a Pareto front technique. This fixes the instability and gives good performance with fully-flexible aeroelastic models.

**Nomenclature**

| | | | | |
|---|---|---|---|---|
| $A$ | state equation matrix; constant | | $\theta$ | generic angle |
| $a$ | generic parameter; axial induction factor | | $\sigma$ | standard deviation |
| $B$ | state equation matrix | | $\tau$ | time constant |
| $b$ | constant | | $\overline{\Psi}$ | integrated speed error |
| $C$ | state equation matrix | | $\overline{\Psi}'$ | weighted integrated speed error |
| $C_P$ | power coefficient | | $\Omega$ | rotor speed |
| $c$ | constant; chord | | $\overline{\Omega}$ | filtered rotor speed |
| $D$ | denominator; output equation matrix; diameter | | $\omega$ | frequency in rad/s |
| $d$ | Dunne's gain factor | | | |
| $F$ | force | | | |



| $f$ | frequency in Hz | | Subscripts: |
|---|---|---|---|
| $H_s$ | significant wave height | 0 | baseline; mean; multi-blade collective |
| $I$ | turbulence intensity | $a$ | aerodynamic |
| $K$ | gain | $c$ | control |
| $L$ | left-hand side state equation matrix; length | DW | dynamic wake mode |
| $M$ | moment | $d$ | driveshaft |
| $N$ | numerator; number of | $e$ | edgewise; element |
| $P$ | power | $F$ | fore-aft |
| $q$ | generalized coordinate | $f$ | flapwise |
| $R$ | outer radius | $I$ | integral |
| $r$ | radius | $i$ | induced |
| $S$ | spectrum | $L$ | low-pass |
| $s$ | Laplace variable | $n$ | natural |
| $T$ | torque | $P$ | proportional |
| $T_p$ | ocean wave peak period | $q$ | quasi-steady |
| $u$ | windspeed fluctuation; generic input | RSC | rotor speed control mode |
| $V$ | windspeed; velocity | $r$ | rated |
| $V_\infty$ | undisturbed windspeed | $S$ | side-to-side |
| $v$ | velocity | $s$ | structural; states |
| $\hat{v}$ | intermediate dynamic wake variable | $u$ | windspeed fluctuation |
| $x$ | states | $x$ | x-direction |
| $y$ | generic output | $y$ | y-direction |
| $\alpha$ | factor on gains; angle-of-attack | $\alpha$ | blade pitch acceleration |
| $\beta$ | blade pitch angle | $\beta$ | blade pitch angle |
| $\zeta$ | damping ratio | $\Omega$ | rotor speed |

## 1 Introduction

It is often taken for granted that a one- or two-degree-of-freedom drivetrain model, with pole-placement techniques, will provide a reasonable gain tuning for the controller of a wind turbine. Hansen *et al.* (2005) describe such a gain tuning and scheduling approach where the target for the rotor speed control mode – that is, the mode which appears above the rated windspeed, where pitching of the blades is used to hold the rotor speed near a constant target value – has a natural frequency





of 0.1 Hz and damping ratio of 0.66. The gains are scheduled on the basis of a single parameter: $\partial P_a/\partial\beta$, the sensitivity of the aerodynamic power with respect to the collective blade pitch angle. This same approach was adopted by Jonkman *et al.* (2009) in the design of the control system for the ubiquitous NREL 5 MW reference wind turbine, which is still the baseline for much of the research on utility-scale wind energy systems.

The flexibility and aerodynamic response of a real wind turbine have a strong influence on the rotor speed control mode. If the gains are not adapted accordingly, the actual mode will have a response which differs significantly from the target frequency and damping ratio. This fact is well-known, also to those authors who have employed the simplistic gain-tuning approaches. The performance of the wind turbine is subsequently verified by aeroelastic simulations, and if the control system performs reasonably from an engineering standpoint, it may be considered a successful design.

There are problems with this approach, though. One- or two-degree-of-freedom models provide little insight into the true dynamics of the system: essentially, the controller tuning is being conducted blindly. The resulting behavior of the rotor speed control mode depends strongly upon the properties of the aeroelastic model, so the same controller may function well or poorly, in a given application. Users of the controller may not understand, or acknowledge, the limitations.

For example, the NREL 5 MW proportional-integral (PI) controller is often adopted as the baseline for comparison against
advanced control algorithms: Schlipf *et al.* (2013), Spencer *et al.* (2013), Jafarnejadsani *et al.* (2014), and Yang *et al.* (2015), to pick some recent examples. Yet this controller is unstable when paired with a fully-flexible aeroelastic model which includes elastic twisting of the blades. For a fair comparison, the reference PI controller should be tuned according to the same model and criteria that were used to demonstrate the performance of the optimal control algorithms; failure to do so weakens the scientific basis of the results.

Controller tuning does not need to be based on a reduced model. Tibaldi *et al.* (2012) optimized the gains of blade pitch and generator torque controllers using full aeroelastic load simulations, combined with a component cost model. The optimization, which ran through seven iterations, was noted to require 4,000 hours computing time, which limits practicability of the method. Nonetheless, the approach of Tibaldi *et al.*, considering the influence of loads and energy production on lifetime cost, is the proper way to evaluate the overall performance of a wind turbine control system.

A practical model for control design contains a minimal number of degrees-of-freedom. It is reasonable to use a low-fidelity model, since a well-designed controller will be robust to small inaccuracies associated with neglected higher-order effects. The model must be of sufficient resolution to capture the important first-order effects. In particular, the frequencies and damping ratios of the control-dominated closed-loop modes within the full model should be preserved in the simple model.

There is not a perfect concensus on which degrees-of-freedom must be included in a model for control design. Among older
publications, Leithead and Connor (2000) is a good place to start, as they conclusively demonstrated that rigid-body models of the drivetrain are inadequate. They included the response of the rotor aerodynamic torque to perturbations in the rotational speed, blade pitch angle, and rotor-average windspeed: $\partial T_a/\partial\Omega$, $\partial T_a/\partial\beta$, and $\partial T_a/\partial u$, respectively. Generator dynamics were also included, as third-order transfer functions, but the blades and tower were considered as rigid. Bossanyi (2000), without providing a formal justification, listed the minimal degrees-of-freedom for design of the blade pitch




controller. The list includes rotor rotation, drivetrain torsion, and tower fore-aft motion as the structural degrees-of-freedom; flexibility of the blades was omitted. Like Leithead and Connor, Bossanyi recommended including generator dynamics, but also added pitch actuator and speed sensor dynamics.

Wright (2004) conducted a methodical investigation into the structural degrees-of-freedom necessary to obtain a stable

control tuning. The 600 kW, 42.6 m diameter CART (Controls Advanced Research Turbine) was used as a reference case. A brief, initial investigation demonstrated the importance of drivetrain flexibility and actuator dynamics on a reference PI controller. A more extensive degree-of-freedom study was conducted with a disturbance-accommodating control (DAC) strategy, which is a state-feedback control algorithm where additional states are used to model, and eventually cancel, disturbances such as turbulence. Though DAC and PI controllers are not identical, lessons learned about the influence of

structural flexibility on a DAC controller can likely be applied to PI tuning as well.

Wright progressively activated one structural degree-of-freedom at a time: drivetrain torsion, collective blade flap, and tower fore-aft. For each set of active degrees-of-freedom, a controller was synthesized, and subsequently evaluated by a brief time-domain simulation with a stepped windspeed profile. It was found that the first blade flapwise modes have the potential to destabilize the first drivetrain mode, and must be included in models for control design. The tower modes, principally the

first fore-aft mode, were found not to have a significant influence on the behavior of the rotor speed control.

Sønderby and Hansen (2014) revisited the question of which degrees-of-freedom should be included in a control-tuning model, in the context of an onshore version of the NREL 5 MW wind turbine. The controller was not specified; rather, the investigation was based on open-loop transfer functions between the actuated degrees-of-freedom – collective blade pitch and generator torque – and generator speed. Particular emphasis was placed on how the poles (indicating frequency and

damping properties) associated with the structural modes changed with the activated structural and aerodynamic degrees-of-freedom. Capturing the non-minimum phase zeros[1] associated with the first tower modes was also of importance.

In contrast with Wright, Sønderby and Hansen found that the control design model should include blade flapwise and edgewise modes, as well as tower fore-aft and side-to-side modes. Quasi-steady aerodynamics may be used at low frequencies (below the first tower modes), though blade torsional flexibility should be included when linearizing the

aerodynamic forces.

The discrepancy between the conclusions of Wright and Sønderby and Hansen can partly be attributed to the increased flexibility of large, multi-MW wind turbines, in comparison with the older CART turbine. However, it is also the case that Sønderby and Hansen, in selecting degrees-of-freedom, applied criteria which were too rigorous, in the context of tuning a

---

[1] Define an actuator-to-output transfer function $y/u = N(s)/D(s)$. A non-minimum phase zero is a factor of the numerator polynomial of the form $(s - a)$. In physical terms, this means that the derivative of the control signal will influence the system in the opposite direction of the control signal itself. In the simple case where $N(s) = s - a$ a step input for $u$ would act as an initial impulse response in the "wrong" direction (associated with the derivative of the step), with a concurrent step response in the "right" direction. Intuitively, it can be appreciated that this limits the frequency band over which the control $u$ can be effective.





typical PI blade pitch controller. Though the poles associated with the structural modes are of concern, as are non-minimum phase zeros, it is not the case that the resulting controller performance is sensitive to each pole, and each zero.

Thus, there is the need to revisit simple models for the design and tuning of PI controllers for highly flexible offshore wind turbines. This is addressed in Section 2, where a study like that of Wright is conducted, incrementally adding degrees-of-

freedom. In contrast with Wright, focus is placed on a PI blade pitch controller, which is evaluated in terms of the pole (frequency and damping ratio) associated with the rotor speed-control mode. A simplified method is also implemented to account for the dynamic wake effect, which may be relevant when gains are low. For basic controller tuning, it is recommended to use, at minimum, a model with elastic driveshaft, blade flap and torsion, and tower fore-aft modes, as well as a dynamic wake.

Since the NREL 5 MW wind turbine controller is used as a baseline in so many studies, including the present one, it is critical to characterize its performance. Dunne *et al.* (2016) have recently found that the effective behavior of this controller is not what one would expect from the reported $K_P$ and $K_I$ gains. The culprit is the scheduling of the integral gain term. Section 3 reviews the gain scheduling of the NREL 5 MW controller in a critical light, and augments Dunne's results with a more revealing physical explanation. It is demonstrated that the controller is unstable near the rated windspeed, when

implemented with a fully-flexible aeroelastic model.

Ocean waves may excite tower resonant vibrations; Section 4 shows that these may appear in the primary blade pitch control path, which is not desireable. In the context of the NREL 5 MW controller, this constrains the cutoff frequency of the low-pass filter on the shaft speed measurement.

The NREL 5 MW wind turbine controller is retuned in Section 6, according to simple metrics of system performance

established in Section 5. The selected tuning is compared against a revised pole-placement approach, using an appropriate aeroelastic model. The fundamental, unavoidable tradeoff is between the fluctuations in rotor speed and the pitch activity; a Pareto front illustrates this explicitly.

The most important finding, deserving of special emphasis, is this: *the appropriate controller tuning is highly dependent on the aeroelastic model*; therefore, no single reference controller can be defined, for use with all models.

## 2 What is the appropriate model fidelity?

Consider the NREL 5 MW wind turbine, mounted atop the OC3 monopile foundation, as described by Jonkman and Musial (2010). The wind turbine is operating in its steady-state condition in a uniform, above-rated windspeed. Fluctuations in the windspeed are, at present, limited to small perturbations about the mean. In this case, the rotor speed and generator power output are controlled as shown in Fig. 1. (All speeds are referred to the low-speed shaft. The gain scheduling in Fig. 1

differs from the controller described by Jonkman *et al.* (2009), for reasons which are made clear in Section 3.)

It is desired to ask some basic questions about this controller. How well does it perform? Could the gains and low-pass filter be chosen differently, to improve the performance? Is the same controller tuning also applicable for an offshore wind



turbine? These are topics of Sections 4 through 6. In order to arrive at the answers, a model of the closed-loop system dynamics is needed. This could be a high-resolution model. Yet there are advantages in adopting a simple model. A simple model is computationally efficient, aids understanding of the system behavior, and can form the basis for more advanced state-space control algorithms. In light of inconsistencies in the literature regarding which degrees-of-freedom are needed, it
is worthwhile to establish some minimum requirements for a model of the closed-loop system dynamics.

## 2.1 The rotor speed control mode

With use of a multi-blade coordinate transform, a three-bladed wind turbine operating under normal conditions (balanced rotor, no extreme excursions) can be represented as a linear time-invariant system, with state and output equations of the form

$$\mathbf{L}\frac{d\mathbf{x}}{dt} = \mathbf{A}\mathbf{x} + \mathbf{B}\mathbf{u} \quad \text{and} \quad \mathbf{y} = \mathbf{C}\mathbf{x} + \mathbf{D}\mathbf{u}. \qquad (1)$$

Hansen (2004) and van Engelen and Braam (2004) describe programs which model wind turbines in this manner; van Engelen and van der Tempel (2004), Merz (2015a), and Tibaldi *et al.* (2015) have extended the scope of linear state-space analysis to the computation of loads under turbulent wind conditions. The present results are obtained using the wind turbine module of the STAS program, which is documented in a series of technical memos (Merz 2015b,c,d). The approach is broadly similar to that of Hansen or van Engelen, and does not warrant a detailed presentation here.

In the discussion that follows we must distinguish between two categories of modes. STAS employs modal reduction of each body (tower, nacelle, driveshaft, and the three blades) prior to assembling the bodies, via constraint equations, into the full wind turbine. For instance, the amplitudes of the first fore-aft and side-to-side modes of the tower body (including the foundation and soil *p-y* springs), are denoted $q_F$ and $q_S$, respectively. These body modes are degrees-of-freedom in the equations of motion; they are elements in the state vector $\mathbf{x}$, as are their time derivatives $dq_F/dt$ and $dq_S/dt$. The body
modes may incorporate features such as bend-twist coupling of the blades.

The second class of modes are the eigenvectors of the equations of motion of the assembled structure, including systems such as the generator, pitch actuators, and controls. These system modes may be dominated by one body mode – for instance, there is an obvious "first tower fore-aft" system mode – or they may have complicated shapes which are not so easily described.

Representing the wind turbine in the form of Eq. (1), the modal properties of the system can be computed. Examination of the system modes reveals one primary and one secondary mode which, within reasonable bounds of the gain tuning, contain the dominant action of the controller. The primary mode can be called the "rotor speed control" mode, as it represents the fluctuation in the rotational speed of the wind turbine rotor, under the combined control actions of the generator and blade pitch actuators. The secondary mode is associated with the influence of dynamic wake effects on the rotor speed control;
this will be called the "dynamic wake" mode. It is most active when control gains are set to comparatively low values.





There is overlap between the rotor speed control and dynamic wake modes. The "rotor speed control" mode contains the dominant rotor speed and blade pitch responses, but the states associated with the dynamic wake – the induced velocities – also participate. The "dynamic wake" mode contains the dominant response of the rotor-wide collective induced velocities, but these are driven by changes in the rotor speed and blade pitch, which also appear in this mode. Thus the *participation of*

*the dynamic wake in the rotor speed control mode* is not to be confused with the *influence of the dynamic wake mode on the rotor speed and blade pitch response*. The former is a dominant effect which is addressed in Section 2.2. The latter, it will be shown shortly, is not so relevant, except when control gains are lower than usual. The salient point is that a dynamic wake model may be needed, even if the "dynamic wake" mode makes little contribution to the response of the relevant control variables.

The rotor speed control mode is clearly visible in transfer functions between axial windspeed and rotor speed. Figure 2 shows these transfer functions, as well as those for blade pitch, at four windspeeds between rated and cut-out. These results were obtained for a full (ca. 600 states) model of the NREL 5 MW wind turbine on a flexible tower and foundation, including soil flexibility.

Figure 2 also lists the natural frequency and damping ratio of the rotor speed control mode. The natural frequency is

associated with the peak in the rotor speed transfer function, while the damping ratio indicates to some extent the sharpness of the peak. Although the rotor speed control mode is dominant, several other system modes also participate in the response. To keep things simple, the discussion of model fidelity is focused on the two system modes with the greatest contribution to the low-frequency rotor speed response. For the baseline gains of Fig. 1, typical participation factors (Kundur 1994) associated with the rotor rotational degree-of-freedom are 0.5 for the dominant rotor speed control mode, and 0.2 for the

secondary dynamic wake mode. These two modes serve as surrogates for the full transfer function: the properties of the transfer function, within the region influenced by the control tuning, can be inferred from the properties of the modes.

As an example, let the NREL 5 MW turbine, on the OC3 monopile foundation, be operating at a mean windspeed of 16 m/s. The baseline gains from Fig. 1 are now modified by a factor,

$$K_P = \alpha K_{P0} \quad \text{and} \quad K_I = \alpha K_{I0}. \tag{2}$$

Figure 3 plots the transfer functions of blade pitch, rotor speed, and tower mudline bending moments, with respect to a

uniform fluctuation in the axial windspeed. The frequency and damping properties of the rotor speed control and dynamic wake modes are tabulated as a function of the gain multiple. At high gains, the peak in the rotor speed transfer function is dominated by the rotor speed control mode, while at low gains, both the rotor speed control and dynamic wake modes make significant contributions.

There is evidently a minimum in the peak sensitivity of rotor speed to fluctuating winds. At high gains, the blade pitch

responds aggressively, in a manner which reduces the damping of the rotor speed control mode; while at low gains, the blade pitch response is so passive that it does not promptly arrest perturbations to the rotor speed.

Within reasonable bounds, gain tuning has little influence on the resonant response of the tower. This is mainly due to the non-minimum phase zero at 0.236 Hz. The presence of this zero is associated with the first tower fore-aft body mode. The





nacelle moves in such a manner that the measured fluctuation in shaft speed is near zero, and there is thus no control response. The particular characteristics of the zero are influenced by other body modes, as well as where in the drivetrain the shaft speed is measured. In the most basic case where the only elastic degree-of-freedom is the tower fore-aft motion, the zero is caused by nacelle fore-aft motion which nearly cancels the fluctuating windspeed. When all the elastic degrees-of-

freedom are included, the motion at the frequency of the zero defies such a simple description, but the outcome is similar. The controller influence at higher frequencies is suppressed by the low-pass filter, with a corner frequency of 0.25 Hz.

The response of the wind turbine depends on both the input-output transfer functions, as in Fig. 2, and the characteristics of the environmental inputs. Typical spectra of rotationally-sampled atmospheric turbulence (the collective component at an outboard blade station) and ocean wave forces are plotted at the left side of Fig. 4. Most of the energy in the turbulence is

concentrated at low frequencies, while that of the ocean waves is in the vicinity of $1/T_p$.

The right-hand side of Fig. 4 shows spectra of the tower mudline bending moments, for three values of the gain multiple α. The peak in the response at low gains is due to the greater energy in the turbulence at low frequencies, while the peak at high gains is due to reduced damping of the rotor speed control mode. (The peak at a gain multiple of 1.5 is not caused by interaction between the controller and ocean waves; Section 4 contains further discussion on this point.) In the present

example, the baseline gains find a happy middle ground. Though not visible in the figure, the response spectra above 0.3 Hz are essentially unaffected by the choice of gains.

To sum up: if we know the natural frequency and damping ratio of the rotor speed control and dynamic wake modes, we can infer much about the response of the wind turbine to the control actions. For a reduced model to be useful in tuning gains, a minimal requirement is that it is able to correctly predict the properties of the rotor speed control mode. If low gains are to

be evaluated – for instance, if the rotor speed control mode might be placed below the ocean wave frequency band – then it is also needed to predict the properties of the dynamic wake mode.

The above statements are valid in the context of basic rotor speed control, for a wind turbine operating above the rated windspeed. Additional control functions – say, active damping of tower or drivetrain resonance – may require that additional system modes are also correctly predicted.

**2.2 The importance of transient aerodynamic loads**

Aerodynamic forces on the blades are subject to transients as conditions change, with a particularly strong effect associated with the blade pitch angle. The transients can be grouped into the categories of circulation lag (Theodorsen), associated with the development of lift along the blade; dynamic stall, connected with movement of the chordwise location of flow separation; and dynamic wake (or dynamic inflow), related to the downstream convection of vorticity in the wake, which

governs the induced velocity at the rotorplane. In an analysis with the blade element momentum method, these phenomena can be represented by a set of linear differential equations, associated with each blade element. The equations employed here are based on the circulation lag method described by Leishman (2002), also Hansen et al. (2004); the Merz *et al.* (2012) variant of the Øye (1990) dynamic stall model; and Øye's dynamic wake model, as documented by Snel and Schepers



(1995). Neglecting the tangential component of induced velocity, the aerodynamic state equations associated with a given blade element are

$$
\frac{d}{dt}
\begin{bmatrix} \hat{v}_i \\ v_i \\ \alpha \\ a_1 \\ a_2 \end{bmatrix}
=
\begin{bmatrix}
-\tau_1^{-1} & 0 & 0 & 0 & 0 \\
\tau_2^{-1} & \tau_2^{-1} & 0 & 0 & 0 \\
0 & 0 & -\tau^{-1} & \tau^{-1}K_1 & \tau^{-1}K_2 \\
0 & 0 & 0 & 0 & 1 \\
0 & 0 & 0 & A_{54} & A_{55}
\end{bmatrix}
\begin{bmatrix} \hat{v}_i \\ v_i \\ \alpha \\ a_1 \\ a_2 \end{bmatrix}
+
\begin{bmatrix}
0.4\tau_1^{-1} & 0 \\
0.6\tau_2^{-1} & 0 \\
0 & \tau^{-1}K_3 \\
0 & 0 \\
0 & 1
\end{bmatrix}
\begin{bmatrix} v_{iq} \\ \alpha_q \end{bmatrix},
\tag{3}
$$

with

$$
A_{54} = -b_1 b_2 \left(\frac{2V}{c}\right)^2, \quad A_{55} = -(b_1 + b_2)\left(\frac{2V}{c}\right), \quad K_1 = (A_1 + A_2)b_1 b_2 \left(\frac{2V}{c}\right)^2,
$$

$$
K_2 = (A_1 b_1 + A_2 b_2)\left(\frac{2V}{c}\right), \quad K_3 = (1 - A_1 - A_2), \quad A_1 = 0.165, \quad A_2 = 0.335, \quad b_1 = 0.0455,
$$

5    $b_2 = 0.3, \quad \tau_1 = \frac{1.1}{1 - 1.3a}\left(\frac{D}{2V_\infty}\right), \quad a = \frac{v_i}{V_\infty}, \quad \tau_2 = \left[0.39 - 0.26\left(\frac{2r}{D}\right)^2\right], \text{ and } \tau = 4.3\frac{c}{V}.$

Examination of the **A** matrix in Eq. (3) shows that the first two states, which represent the dynamics of the rotor wake, are not directly coupled with the remaining three states, which represent the circulatory flow local to the airfoil. The "dynamic wake" and "circulation" effects are interdependent, when linked into the full state-space model of the wind turbine; but they can be independently activated or deactivated.

10    Considering first the dynamic wake, Fig. 5 shows transfer functions of rotor speed and blade pitch with respect to rotor-average windspeed. The solid curves were computed in the frequency domain, based upon a linear state-space model. Two cases are shown, one with the dynamic wake model active, and another with it inactive, such that the induced velocities are always in equilibrium with the airfoil forces.

Nonlinear time-domain results were obtained by defining a spatially-uniform wind field, whose axial velocity component 15   varied in time about the mean, with a prescribed frequency, and an amplitude of 0.5 m/s. Observing the output rotor speed or blade pitch – after at least 30 seconds to allow startup transients to decay – the amplitude of the signal at the input frequency was extracted. The fundamental frequency was dominant in all cases, except immediately in the vicinity of the tower resonant frequency, where beating was observed; this does not impact the rotor speed control mode. Time-domain results at a windspeed of 12 m/s could not be generated, due to an instability in the controller, which is discussed in Section 20   3.4.

The FAST v8 program (Jonkman and Jonkman 2016), with the BeamDyn blade module and AeroDyn v15 aerodynamic module, was used for the time-domain calculations. This version of AeroDyn included the Beddoes-Leishman model of transient circulation, but it was limited to an equilibrium wake; thus the time-domain results should be compared against the black curves. The FAST model did not include the elastic properties of the seabed; these were stiffened in the linear model, 25   in order that the tower natural frequencies should match. (For other analyses, the seabed properties have been represented by





p-y springs, for a more accurate estimate of the tower natural frequencies, which should be around 0.24 Hz; the difference is negligible, in terms of the behavior of the rotor speed control mode.)

The principal effect of the dynamic wake is that a blade pitch action results in an initially large change, or overshoot, in the airfoil forces. If the pitch angle is subsequently held steady, the forces decay to their quasi-steady values over a timescale of

roughly $D/V_\infty$. In other words, the aerodynamic forces are more sensitive to blade pitch – directly influencing the rotor speed control mode – when the dynamic wake model is active. This is reflected in a higher frequency and lower amplitude of the rotor speed and blade pitch responses.

Above the rated windspeed, mean induced velocities decrease with the windspeed, so the dynamic wake becomes less significant at higher mean windspeeds.

Transient circulation has a moderate influence on rotor speed control, reducing the damping, as illustrated by the transfer functions in Fig. 6.

**2.3 A (very) simple transient aerodynamic loads model for controller tuning**

If one is interested in the bulk flow characteristics across the rotor, as would be relevant for tuning of a collective pitch controller, it is not desireable to retain element-by-element resolution over the span.

A simple method is suggested to "collapse" the transient aerodynamics into a set of equations associated with a single blade element. The transients of state space Eq. (3) are computed according to a representative blade element at $r/R = 0.75$ (or rather, the collective component, which is nothing more than the average over the three blades), and then these characteristic transients are assumed to apply to the sum of the aerodynamic forces over all the blade elements. The choice of $r/R = 0.75$ is simply to obtain a value which is representative of an outboard blade station, not too close to the tip. The results are not

sensitive to the precise radial location.

It is perhaps easiest to explain this operation by sketching the process by which the aerodynamic states are reduced, as in Fig. 7. For simplicity, this is presented as if there were only one aerodynamic state associated with each blade element; the process is identical for each of the five types of states in Eq. (3). In the state vector, there is a group $\mathbf{x}_s$ with the structural states, a group $\mathbf{x}_a$ with the aerodynamic states, and a group $\mathbf{x}_c$ with the control states. The parts shaded gray, associated with

the aerodynamic states, are partitioned out and deleted. Only one aerodynamic state is retained, that associated with the collective component at $r/R = 0.75$. Its row, shown as a black line, is unchanged (apart from the deleted columns). However, in the rows of the $\mathbf{A}$ matrix associated with the structural states, the columns associated with the aerodynamic states (light gray) are summed into the column of the retained aerodynamic state (white bar). In this manner, the transient evolution of the single retained aerodynamic state comes to represent the response over the entire rotor.

(The $\mathbf{L}$ matrix is also modified, but as the only nonzero elements in the relevant rows and columns lie on the diagonal, the operation is trivial.)

The process is repeated for each of the five types of aerodynamic states in Eq. (3), and the result is that five states represent the collective, transient aerodynamics of the rotor.





There are other, more formal methods by which the number of aerodynamic states could be reduced. A low-order series representation of rotor induction, such as the acceleration potential method described by Burton *et al*. (2001), is one possibility. Another is the modal reduction approach of Sønderby (2013); though as derived this did not include a dynamic wake.

Nonetheless, the above ad-hoc matrix reduction method works well for the present purpose of control tuning, where low-frequency, rotor-average wind inputs are of greatest concern. The reduced model is validated, in Table I and Section 5, against the original matrices employing the full radius-dependent Øye model of Eq. (3).

### 2.4 Degrees-of-freedom

A series of models was constructed, progressing from the simplest case with only rigid-body rotation of the rotor, through to
the full case with the elastic structure represented by 110 modal degrees-of-freedom. The reduced models employed either quasi-steady aerodynamics, or the five-state transient model of Section 2.3, whereas the full model employed a blade element momentum method with transients computed for each element. For the reduced models, the number of states $N_s$ consists of one state for rotor rotation, two for each elastic degree-of-freedom, two for the controller, and five for transient aerodynamics. Blade pitch is directly prescribed by the controller. The full model includes additional states describing the
blade pitch response, as well as the electrical dynamics of the generator (Merz 2015d).

Table I lists the models. For each model, the natural frequency and damping ratio of the rotor speed control mode were computed at three above-rated windspeeds: 12, 16, and 20 m/s. Here the baseline gains of Fig. 1, corresponding to $\alpha = 1$ in Eq. (2), were used. At this level of gain, the dynamic wake mode was highly damped, with a damping ratio of nearly unity. To better illustrate this mode, and verify that it was well-predicted where it matters, its properties were computed again using
a reduced gain factor of $\alpha = 0.5$.

The "correct" result is taken to be that obtained with the full linear model, highlighted in bold.

The conclusion is that models which do not include at least tower fore-aft, blade flapwise, and torsional flexibility may give misleading estimates of the rotor speed control response, and are therefore unfit for the purpose of tuning the controller. With blade flap and torsion, tower fore-aft, and a five-state transient aerodynamic model, the rotor speed control and
dynamic wake modes are well-predicted. Adding blade edge and tower side-to-side flexibility makes little difference. Model 5D is therefore recommended as a minimal model for controller gain tuning.

Drivetrain torsional flexibility is expected to have little influence on the basic control tuning, provided that the low-pass filter frequency is reasonably low; however, this degree-of-freedom is retained in the models in deference to common practice. It is indeed important to evaluate the damping of the first drivetrain mode, as this can potentially be destabilized by
the generator torque control. Model 7D is recommended for evaluating the first drivetrain torsional resonance mode, as this is influenced by the flexibility of the blade edgewise and tower side-to-side modes.



It is emphasized that other control functions which are not shown in Fig. 1 may require additional degrees-of-freedom. Incorporating environmental inputs such as misaligned ocean waves may also require additional degrees-of-freedom, at least those of Model 7D.

### 3 Revisiting a baseline control architecture

The nonlinear, gain-scheduled NREL 5 MW wind turbine controller, during operation above the rated windspeed, is shown in Fig. 8. The rate limits, pitch angle limits, and integral gain saturation are omitted. With the exception of the 0° minimum pitch angle, these limits are seldom reached during normal operating conditions. The pitch angle hits 0° when the windspeed dips below the rated value, and the control mode transitions between constant-power and constant tip-speed ratio (maximum $C_P$) operation. This transition is not crucial to the present argument, and is neglected for the time being.

The critical feature to note is that the scheduling of the integral gain happens outside the integral of the speed error, $\overline{\Psi}$. That is, the contribution of the integral pathway to the demanded blade pitch angle is

$$K_I(\beta) \int_0^t (\overline{\Omega} - \Omega_r) \, d\tau. \tag{4}$$

It will be shown that this leads to a misleading definition of proportional and integral gain. An alternative is to schedule the integral gain inside the integral,

$$\int_0^t K_I(\beta)(\overline{\Omega} - \Omega_r) \, d\tau \tag{5}$$

also shown at the right of Fig. 8. In state space, the equations describing the controller are

$$\frac{d}{dt}\begin{bmatrix} \overline{\Psi} \\ \overline{\Omega} \end{bmatrix} = \begin{bmatrix} 0 & 1 \\ 0 & -\omega_L \end{bmatrix} \begin{bmatrix} \overline{\Psi} \\ \overline{\Omega} \end{bmatrix} + \begin{bmatrix} 0 & -1 \\ \omega_L & 0 \end{bmatrix} \begin{bmatrix} \Omega \\ \Omega_r \end{bmatrix} \tag{6a}$$

$$\beta = \begin{bmatrix} K_I(\beta) & K_P(\beta) \end{bmatrix} \begin{bmatrix} \overline{\Psi} \\ \overline{\Omega} \end{bmatrix} + \begin{bmatrix} 0 & -K_P(\beta) \end{bmatrix} \begin{bmatrix} \Omega \\ \Omega_r \end{bmatrix}, \tag{6b}$$

where the gain is outside the integral, versus

$$\frac{d}{dt}\begin{bmatrix} \overline{\Psi'} \\ \overline{\Omega} \end{bmatrix} = \begin{bmatrix} 0 & K_I(\beta) \\ 0 & -\omega_L \end{bmatrix} \begin{bmatrix} \overline{\Psi'} \\ \overline{\Omega} \end{bmatrix} + \begin{bmatrix} 0 & -K_I(\beta) \\ \omega_L & 0 \end{bmatrix} \begin{bmatrix} \Omega \\ \Omega_r \end{bmatrix} \tag{7a}$$

$$\beta = \begin{bmatrix} 1 & K_P(\beta) \end{bmatrix} \begin{bmatrix} \overline{\Psi'} \\ \overline{\Omega} \end{bmatrix} + \begin{bmatrix} 0 & -K_P(\beta) \end{bmatrix} \begin{bmatrix} \Omega \\ \Omega_r \end{bmatrix}, \tag{7b}$$

where the gain is now inside the integral. A prime is added to $\overline{\Psi'}$, as this is not the same as the integrated speed error $\overline{\Psi}$.

Dunne *et al.* (2016) identified the fact that by scheduling the integral gain outside of the accumulated speed error, the effective gains, for small perturbations about a mean operating point $\beta_0$, are significantly lower than $K_P(\beta)$ and $K_I(\beta)$. Here this finding is given a deeper physical explanation.





### 3.1 The role of the integral pathway

For a steady-state operating point, with zero speed error, the integral pathway provides the mean blade pitch angle set-point. This is clearly illustrated by observing the behavior of the controller while the turbine starts up in a condition of above-rated windspeed; Fig. 9 is an example. In this simulation, using the FAST v8 program, the windspeed was a constant 15 m/s and

the structure was rigid. The initial conditions were a rotor speed equal to the rated speed of 12.1 rpm (in order to avoid implementing specific startup control logic), a blade pitch angle of 0°, and a low-pass filtered speed error and integrated speed error of zero. The full nonlinear NREL 5 MW controller was employed, including limits, although these did not come into effect.

Thus, the integrated error $\overline{\Psi}$ has a comparatively large mean offset, when the turbine is operating above the rated

windspeed.

The integral pathway also acts, together with the low-pass filter, to determine the lag between fluctuations in the rotational speed and the blade pitch angle, which in turn influences the response of the system. This is best illustrated in the frequency domain, as in the following section, where the concept of phase can be applied.

### 3.2 The effect of scheduling the integral gain

Let the NREL 5 MW wind turbine be operating in a uniform, steady, above-rated wind. Let there be a small perturbation to the shaft speed, $\Delta\Omega$, which could be the result of, say, a small fluctuation in the windspeed. The state equation of the baseline controller, Eq. (6a), is linear and therefore of the same form,

$$\frac{d}{dt}\begin{bmatrix}\Delta\overline{\Psi}\\\Delta\overline{\Omega}\end{bmatrix} = \begin{bmatrix}0 & 1\\0 & -\omega_L\end{bmatrix}\begin{bmatrix}\Delta\overline{\Psi}\\\Delta\overline{\Omega}\end{bmatrix} + \begin{bmatrix}0\\\omega_L\end{bmatrix}\Delta\Omega , \tag{8a}$$

for small perturbations. Note that $\Omega_r$ is constant, and so vanishes from the perturbation equations. Linearization of the nonlinear gain-scheduled blade pitch output, Eq. (6b), gives

$$\Delta\beta = \left(1 - \frac{\partial K_I}{\partial\beta}\bigg|_0 \overline{\Psi}_0\right)^{-1}\begin{bmatrix}K_{I0} & K_{P0}\end{bmatrix}\begin{bmatrix}\Delta\overline{\Psi}\\\Delta\overline{\Omega}\end{bmatrix}. \tag{8b}$$

As $\partial K_I/\partial\beta|_0$ is a negative value, and $\overline{\Psi}_0$ is a positive value, the effective gains which multiply $\Delta\overline{\Psi}$ and $\Delta\overline{\Omega}$ are smaller than the respective $K_{I0}$ and $K_{P0}$. By contrast, linearization of the controller of Eq. (7a) and (7b), with the gain scheduling inside the integral, gives

$$\frac{d}{dt}\begin{bmatrix}\Delta\overline{\Psi}\\\Delta\overline{\Omega}\end{bmatrix} = \begin{bmatrix}0 & 1\\0 & -\omega_L\end{bmatrix}\begin{bmatrix}\Delta\overline{\Psi}\\\Delta\overline{\Omega}\end{bmatrix} + \begin{bmatrix}0\\\omega_L\end{bmatrix}\Delta\Omega \tag{9a}$$

$$\Delta\beta = \begin{bmatrix}K_{I0} & K_{P0}\end{bmatrix}\begin{bmatrix}\Delta\overline{\Psi}\\\Delta\overline{\Omega}\end{bmatrix}, \tag{9b}$$

after rearranging to replace $\overline{\Psi}'$ with $\overline{\Psi}$. It is now argued that Equations 7a and 7b, with linearizations 9a and 9b, are unambiguously the correct way to define the behavior of a PI control system. If one looks at the signals coming from the





proportional and integral pathways, for small perturbations about the mean, Equation 9b (scheduling inside the integral) gives

$$\Delta\beta_P = K_{P0}\Delta\overline{\Omega} \quad \text{and} \quad \Delta\beta_I = K_{I0}\Delta\overline{\Psi}, \quad \text{with} \quad \Delta\beta = \Delta\beta_P + \Delta\beta_I, \tag{10}$$

which is exactly what is expected. On the other hand, Equation 8b (scheduling outside the integral) results in

$$\Delta\beta_P = K_{P0}\Delta\overline{\Omega} \quad \text{and}$$

$$\Delta\beta_I = \left(1 - \frac{\partial K_I}{\partial\beta}\bigg|_0 \overline{\Psi}_0\right)^{-1} K_{I0}\Delta\overline{\Psi} + \left[\left(1 - \frac{\partial K_I}{\partial\beta}\bigg|_0 \overline{\Psi}_0\right)^{-1} - 1\right] K_{P0}\Delta\overline{\Omega}. \tag{11}$$

That is, the signal coming from the integral pathway contributes both integral and proportional effects. This is confusing, to

say the least.

The behavior of the two versions of the controller can be visualized in the frequency domain, using phasors, as shown in Fig. 10. This particular phasor diagram was generated using Model 5Q of Section 2.4, during normal operation at a mean windspeed of 15 m/s, for a unit shaft speed input. The diagram varies with frequency; here 0.1 Hz, the design frequency for the rotor speed control mode, is shown. The figure is qualitatively the same for any frequency which is well below that of

the first tower mode.

In the present example, all quantities are referred to the low-speed shaft, with $K_{P0} = 0.695$ s, $K_{I0} = 0.296$, $\partial K_I/\partial\beta = -1.03$, and $\overline{\Psi}_0 = 0.6$ rad.

The phasor diagram is interpreted as follows. The low-pass filter on the shaft speed fluctuation $\Delta\Omega$ gives a delayed, and slightly suppressed, measured speed $\Delta\overline{\Omega}$; the phase lag is a function of the input and low-pass filter frequencies. By

definition, the integral of the measured speed, $\Delta\overline{\Psi}$, lags the measured speed by 90°. The signal $\Delta\beta_P$ through the proportional pathway is, in both cases, $K_{P0}\Delta\overline{\Omega}$. However, the signal $\Delta\beta_I$ through the integral pathway is, for gain scheduling outside the integral, computed by Eq. (11). This gives a phase lag with respect to $\Delta\overline{\Psi}$, which reduces both the effective proportional and integral gains, according to Eq. (8b).

### 3.3 Effective gains of the NREL 5 MW controller

Comparing Equations 8b and 9b, it is evident that when the gain is scheduled outside the integral term, the controller behaves as though the baseline gains $K_{I0}$ and $K_{P0}$ are reduced by the factor

$$d := \left(1 - \frac{\partial K_I}{\partial\beta}\bigg|_0 \overline{\Psi}_0\right)^{-1}, \tag{12}$$

where $d$ stands for "Dunne's gain factor". For the NREL 5 MW controller, this is the curve shown in Fig. 11, when plotted against the mean windspeed.

The gain factor $d$ makes a big difference. Fig. 12 plots the natural frequency and damping ratio of the rotor speed control

mode, with and without the factor. These results were generated using the full (572-state) linear model of Table I, including transient aerodynamic effects.



### 3.4 Instability of the NREL 5 MW controller

According to Fig. 12, the rotor speed control mode of the NREL 5 MW turbine, with its baseline controller, is unstable in the vicinity of the rated windspeed. Whether the instability is present in a given analysis depends upon the degrees-of-freedom implemented in the aeroelastic model. Blade torsional flexibility is of particular importance; at a windspeed of 11.5 m/s,

Model 6D (blade stiff in torsion) predicts a damping ratio of +0.296, whereas Models 7D and 8D (blade flexible in torsion) predict +0.012 and -0.019, respectively.

The instability is confined to a narrow range of operation. On the low-windspeed side, it is bounded by the control mode transition from rated power and speed, to maximum $C_P$ tracking. On the high-windspeed side, the gain scheduling provides stable operation.

Yet, the instability is significant. Near the rated windspeed, the controller is driven through a greater number of mode transitions than necessary, which leads to more variability in the power production. The blade pitch is more active than necessary, which is reflected in both the pitch actuator duty cycle and the fluctuating loads on the blades, drivetrain, and support structure.

The instability can be demonstrated in the time domain. Fig. 13 shows the response to a uniform wind which decreases in

steps, at intervals of 30 s, from 12.5 m/s to 11.6 m/s. The analysis was performed using FAST v8/BeamDyn, which includes blade torsional flexibility.

It is concluded that the baseline NREL 5 MW controller is workable, if the blades are modelled as rigid in torsion; but only because the inaccuracies associated with the simple rigid-shaft model used for gain tuning were counterbalanced by the effect of scheduling the gains outside the integral. If run with a fully flexible model, the baseline controller is unstable in an

interval just above the rated windspeed. As a consequence, the many wind turbine control studies which have used the NREL 5 MW controller as a baseline have been comparing against a PI controller whose tuning is somewhat arbitrary.

The essence of this conclusion is not unique to the NREL 5 MW controller. *The appropriate controller tuning is highly dependent on the aeroelastic model*; therefore, no single reference controller can be used with all models.

### 4 The influence of ocean waves on control actions

Ocean waves excite tower motions, and this can influence the rotor speed and control actions. As seen in Fig. 1, the cutoff frequency of the low-pass filter frequency of the NREL 5 MW wind turbine controller is 0.25 Hz, which is above the wave frequency band, and nearly the same as the first tower natural frequency. This means that the controller will respond to wave-driven motions of the structure, if these perturb the rotor speed measurement.

There are two ways in which structural motion could influence the rotor speed measurement. One is via a change in the

relative windspeed, and thus the aerodynamic forces on the rotor. The other is by causing rotation of the nacelle about the axis of the driveshaft. A speed measurement at the low-speed shaft, as on a direct-drive wind turbine, is particularly susceptible to the latter effect.



The influence of ocean waves was determined by examining the closed-loop transfer functions between waterline wave forces and rotor speed, $\partial\Omega/\partial F_x$ and $\partial\Omega/\partial F_y$, where $x$ is is the along-wind direction and $y$ the cross-wind direction. For this purpose, a direct-drive version of the 5 MW turbine nacelle was modelled, so the speed measurement is on the low-speed shaft.

Figure 14 plots the transfer functions, and also the spectra of rotor speed fluctuations, for ocean waves with $H_s = 2$ m and $T_p = 6$ s.

To avoid the situation where the controller responds to wave-driven tower resonance, it is recommended to set the low-pass filter cutoff frequency to a value well below the first natural frequency of the tower. It is acceptable – and likely unavoidable – that the cutoff frequency then lies within the wave-frequency band of roughly 0.05-0.20 Hz.

While it is desired to minimize the response of the primary PI control path to tower motions, this does not rule out active damping of the tower. Active damping can be implemented via an auxiliary control path, where the phase is adjusted to maximize the damping effect.

## 5 Metrics for evaluating control performance

When tuning the gains and filters of a wind turbine controller, it makes sense to implement some indicative performance
metrics, in addition to stability criteria. The reason is that the environmental load inputs are highly nonuniform, in terms of the spectra, or frequency content of the signals; Fig. 4 illustrates this point. The response of the system depends on the properties of the modes – in particular the rotor speed control mode – in relation to the inputs.

Above the rated windspeed, the primary functions of the controller are to keep the rotor speed near the rated value, and the generator producing the rated power, while preventing the generator from exciting drivetrain torsional resonance. The pitch
actuator duty cycle is also of concern, and the pitch action has a strong influence on the structural response.

A set of simple metrics could then be the standard deviations of the rotor speed, $\sigma_\Omega$, which is the primary control function; the blade pitch acceleration, $\sigma_\alpha$, as this is proportional to the torque delivered by the pitch actuator; and the fore-aft displacement of the nacelle, $\sigma_F$, which is indicative of the internal bending moments in the tower. More elaborate derived quantities such as damage-equivalent loads could also be employed; but it was desired to keep things simple.

In order to compute the stochastic response of the wind turbine using the simplified transient aerodynamic method of Section 2.3, the turbulent wind field must be reduced to a single rotor-average windspeed input. The starting point is the full matrix of rotationally-sampled turbulence cross-spectra between blade elements. Merz *et al.* (2012, 2015c) describe the methods used to generate this spectral matrix. Velocity cross-spectra between each pair of rotating blade elements are computed analytically using isotropic turbulence theory, together with the Von Karman spectrum. The resulting spectral matrix is
transformed into multi-blade coordinates, giving the characteristic $3nP$ signals in the ground-fixed frame.

The collective multi-blade components of the spectral matrix are retained, and the cosine and sine components are discarded. Then an averaging procedure is performed, weighting the contribution at each blade element according to its swept area





($\approx 2\pi r_e L_e$), which is used as a surrogate for how important each radial station is to the rotor loading. The equation for the weighted average is

$$\bar{S}_u(f) = \frac{(\mathbf{L}_e \circ \mathbf{r}_e)^T \mathbf{S}_u(f)(\mathbf{L}_e \circ \mathbf{r}_e)}{\left(\sum_k L_{e,k} r_{e,k}\right)^2}, \tag{13}$$

where $\mathbf{L}_e$ and $\mathbf{r}_e$ are column vectors of the spanwise length and radial coordinate of each blade element, $\mathbf{S}_u$ is the spectral matrix of collective multi-blade components of turbulence, and $\circ$ denotes elementwise multiplication (Hadamard product).

In cases with ocean waves, the wave force spectrum is derived by running a time-domain hydrodynamic analysis, summing the forces to a point on the tower at the waterline, and computing the spectrum from the time-series of forces.

Model 7D, with Eq. (13), is capable of approximating the standard deviation metrics, local to an operating point, from a full linear model of the wind turbine. Figure 15 shows the spectra of rotor speed, nacelle fore-aft displacement, and blade pitch acceleration, for small stochastic perturbations ($I = 0.02$) about nominal operating windspeeds of 16 and 20 m/s. The

standard deviation follows as the square root of the integral under the spectral curve; these are summarized in Table II.

Model 7D, with the single turbulence input of Equation 5.1, is compared against a full (572-state) model, with a full 3D input turbulence field. Corresponding results were also generated with a nonlinear time-domain model. The blade pitch angle was kept in the vicinity of the operating point by the low value of turbulence intensity, such that the influence of gain scheduling was negligible.

Model 7D provides an accurate estimate of the rotor speed and tower fore-aft displacement from the full linear model. The estimate of blade pitch acceleration is not precise, but it is reasonable. The same can be said for the comparison between the linear model and FAST: the agreement is not precise, but it is reasonable, seen in the light of the variability typically encountered in code-to-code comparisons. The trends in the spectra give confidence that the relevant physical phenomena are represented.

There is one exception: tower side-to-side resonance – visible in the rotor speed and blade pitch responses – which is much more pronounced in the nonlinear analysis. This discrepancy is curious, since the fore-aft response matches nicely; it is likely attributable to the low side-to-side damping. At a windspeed of 16 m/s, the linear model predicts a side-to-side damping ratio of 0.0075, whereas the value computed from a decay test, using FAST, was approximately 0.003. In addition, Model 7D does not fully capture the pitch response to 3P turbulence sampling, around 0.6 Hz. A better filter on the speed

measurement, or a more accurate pitch actuator model (nonexistant in the NREL 5 MW reference wind turbine), would likely reduce the significance of the 3P blade pitch response. These items deserve a deeper investigation; but as they do not appreciably impact the present results, this is left to future work.

Though not as crucial to controller tuning at a given operating point, it is also of interest to evaluate how well a linear model can predict stochastic fluctuations under realistic operating conditions. The analyses were repeated with $I = 0.16$, with the

resulting standard deviations listed in Table II, and spectra plotted in Fig. 16. Interestingly, despite the increased amplitude, the degree of agreement between the linear and nonlinear models is essentially unchanged. This hints that modelling





assumptions, rather than fundamental nonlinearities, may play a dominant role in the discrepancy between the linear and nonlinear analyses.

In the simulation with a mean windspeed of 16 m/s, the controller cycled 12 times, in the 2400 s timespan, through the unstable region, with a control mode transition. Since the NREL 5 MW wind turbine includes no pitch actuator dynamics, the saturation of the blade pitch at 0° was essentially instantaneous. This renders the computed pitch accelerations meaningless. Rather, in Fig. 16, the blade pitch angle spectra are compared.

Based on these results, it can be expected that Model 7D provides the correct trends in the performance metrics, and is useful for stability analysis and gain tuning at an operating point.

## 6 Tuning the baseline controller for use with fully-flexible aeroelastic models

The baseline tuning of the NREL 5 MW controller is workable, when used in combination with simplified aeroelastic models which do not include blade torsional flexibility. For more advanced aeroelastic models, a different controller tuning is required in order to eliminate the instability, and improve the overall performance, near the rated windspeed.

The retuning could be as simple as reapplying the pole-placement strategy described in the introduction, using Model 5D (or 7D or 8D) instead of Model R. Selecting $f_L = 0.20$ Hz, for the reasons discussed in Section 4; and maintaining the targets of 0.1 Hz frequency and ~0.6 damping ratio; the gains are then scheduled as

$$K_P = 0.5679 - 3.409\beta + 21.07\beta^2 - 67.78\beta^3 + 74.77\beta^4$$
$$K_I = 0.05417 - 0.5909\beta + 7.454\beta^2 - 24.19\beta^3 + 26.69\beta^4. \tag{14}$$

These values are valid for $0 \leq \beta \leq 0.40$ rad; for transient excursions above 0.40 rad during normal operation, the gains are computed according to $\beta = 0.40$ rad.

An alternative, in the manner of Tibaldi *et al.* (2012), is to tune the controller gains and filters based upon an evaluation of system performance. Here we use the metrics of Section 5, together with Model 7D. This model runs quickly enough that a complete mapping of the tuning parameters, within reasonable bounds, is feasible. Sophisticated optimization techniques are not needed.

As an example of one possible approach for tuning the controller, consider the case with a mean windspeed of 16 m/s. For appropriate weighting of wind and wave loads, a realistic value of the turbulence intensity is selected: the IEC Class IB normal turbulence model gives $I = 0.154$. An ocean wave climate of $H_s = 2$ m and $T_p = 6s$ is representative of the North Sea in the given wind conditions (Fischer *et al.* 2010). The waves were given a misalignment of 15°, in order to excite some tower side-to-side motion, such that this may be reflected in the rotor speed and blade pitch metrics.

Figure 17 plots the Pareto front, in terms of the objectives: minimization of $\sigma_\Omega$, $\sigma_\alpha$, and $\sigma_F$. The same points are plotted twice, at left in terms of the objectives, and at right in terms of the control tuning parameters. Some of the points in the right-hand plot overlap; the highest value of $K_I$ is shown.



The curved lower boundary visible in the left-hand plot represents the fundamental tradeoff between pitch activity and rotor speed; tightly limiting fluctuations in rotor speed requires rapid pitch action, hence high pitch accelerations. Extremes in either direction – very aggressive or very passive control – are associated with more severe structural loads. A balanced tuning is preferred. It is possible to further reduce structural loads by straying from the lower boundary, sacrificing some

performance in terms of pitch activity and rotor speed.

The plot at right illustrates that, among the Pareto-optimal points, a high filter frequency and high proportional gain are associated with a high integral gain; likewise for low values. The trends are summarily explained: cases with a high filter frequency and high gains lie on the Pareto front because they minimize the deviations in rotor speed; and cases with a low filter frequency and low gains minimize the pitch activity. Other cases represent varying degrees of tradeoffs between rotor

speed, pitch activity, and structural response.

To pick one tuning from the Pareto front requires some assumptions; there is no single "correct" solution. Let us propose some guidelines: (1) Comparatively tight speed control and responsive pitch control is desired in the vicinity of the mode transition at the rated windspeed. (2) The metrics are more important near the rated windspeed, where the turbine will be operating most often. (3) In comparison with the pole-placement tuning, we wish to trade a somewhat increased pitch

activity for tighter speed control and reduced structural motions.

Extended to the full range of windspeeds between rated and cutout, these guidelines suggest that the gains

$$K_P = 1.071 - 3.651\beta + 1.666\beta^2 + 16.25\beta^3 - 21.34\beta^4$$
$$K_I = 0.1679 + 0.1771\beta - 12.16\beta^2 + 60.52\beta^3 - 79.61\beta^4$$

(15)

are a good choice. These have the same range of validity as Eq. (14). Figure 18 compares the baseline, pole-placement, and Pareto gains, where the baseline gains include Dunne's gain factor, Eq. (12).

A low-pass filter frequency of 0.17 Hz was found to be reasonable over the entire windspeed range between rated and cutout.

The best metrics, according to the chosen performance criteria, were obtained within $0.17 \leq f_L \leq 0.27$ Hz at windspeeds near rated, shifting gradually to $0.12 \leq f_L \leq 0.17$ Hz near cut-out. This indicates that it could be beneficial to schedule the low-pass filter frequency in addition to the gains. There is no particular difficulty in doing so. At the same time, the observed benefits were minor, and it was decided that these did not justify diverging from the baseline control strategy of a constant low-pass filter frequency – at least, for operation above the rated windspeed. Energy production (maximum $C_P$

tracking) below the rated windspeed was not considered.

Tables III and IV list the gains, together with the primary metrics $\sigma_\Omega$, $\sigma_\alpha$, and $\sigma_F$, used to generate the Pareto front. Standard deviations of other degrees-of-freedom are also shown. These tables were generated with IEC Class IB normal turbulence, and the ocean wave conditions mentioned previously. Tables V and VI list the frequency and damping properties of selected system modes.

The "preferred" damping ratio of the rotor speed control mode is roughly 0.3, in contrast with the value of 0.6 chosen for pole placement. Tighter control of the rotor speed is achieved not by increasing the damping, but rather by increasing the frequency. This moves the peak in the $\partial\Omega/\partial u$ transfer function away from the high-energy, low-frequency turbulence,





giving response spectra as shown in Fig. 19. Note that the magnitude of the pitch angle fluctuations is nearly independent of the tuning; thus the pitch acceleration depends primarily upon the frequency.

The natural frequency of the rotor speed control mode tends to increase at windspeeds approaching cutout. The pole-placement technique, holding the frequency at 0.1 Hz, requires a comparatively high integral gain (Table III), in relation to

the proportional gain – the integral path acts as a negative stiffness on the speed fluctuations.

## 7 Conclusions

There is no single reference control tuning which performs well with all types of wind turbine models. It is therefore incumbent upon the analyst to understand the properties and limitations of the model, and select a control tuning that gives the desired behavior. The aspects of behavior relevant to control tuning can be largely understood in terms of the rotor speed

control mode of the closed-loop system.

Rule-of-thumb methods, using pole placement on a rigid rotor, are inadequate. The NREL 5 MW wind turbine controller was tuned in this manner, and it is unstable near the rated windspeed, when paired with a fully-flexible aeroelastic model. This calls into question some of the comparisons between optimal and PI controllers which have been published over the last decade.

Simple models, with 16-20 states, can be used to tune the controller for use with a fully-flexible aeroelastic model. The degrees-of-freedom must be selected with care: at a minimum, a model for control tuning requires tower fore-aft, driveshaft, blade flapwise, and blade torsional flexibility. A dynamic wake model is also needed.

The control-tuning models can be generated in an automated manner from full linearized models, which are commonly available as output from aeroelastic codes. The matrices are partitioned, retaining only the rows and columns associated

with the selected states. It is acceptable, for purposes of control tuning, to retain only one set of transient aerodynamic equations, which then represents the rotor-average aerodynamic response. The appropriate representation of the turbulent wind spectrum then becomes a challenge. A simple solution is to use the swept-area-weighted average of the collective, rotationally-sampled turbulence components associated with each blade element. The spectral matrix is thereby reduced to a single scalar input.

The proper architecture of a gain-scheduled PI controller places the scheduled integral gain inside the integral of the error.

Ocean waves, especially when misaligned with respect to the wind, drive tower side-to-side vibrations, to which the pitch controller may respond. It is recommended to low-pass filter the shaft speed measurement with a cutoff frequency that is well below the frequency of the first tower fore-aft mode.

A revised controller tuning was developed for the NREL 5 MW wind turbine, reducing the low-pass filter cutoff frequency,

and rescheduling the gains. A Pareto optimization approach was used in order to identify a set of gains which give a stable, balanced performance in terms of minimizing the rotor speed error, the blade pitch accelerations, and the structural motions.





**Acknowledgements**

This work has been funded by NOWITECH, the Norwegian Research Centre for Offshore Wind Technology, www.sintef.no/projectweb/nowitech.

Many thanks to Fiona Dunne of the University of Colorado, Boulder for providing a pre-publication version of a manuscript

addressing the gain scheduling issue of Section 3. Thanks also to Jason Jonkman of NREL for facilitating this exchange.

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



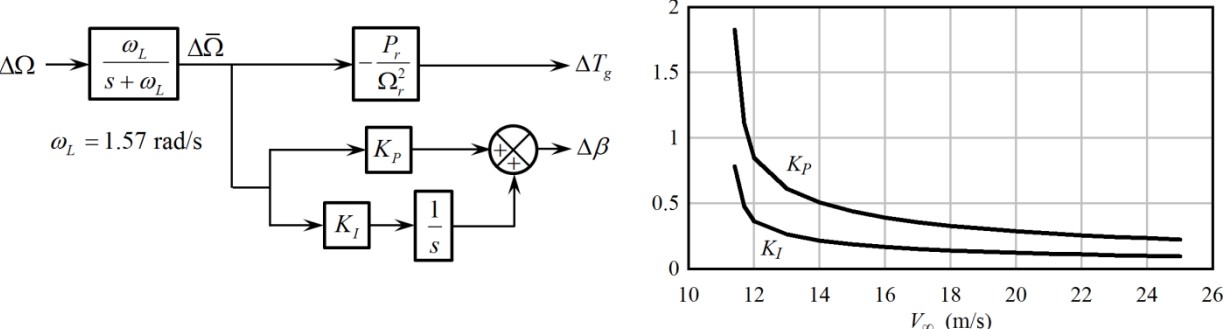

Figure 1: The effective above-rated blade pitch and generator torque controller of the NREL 5 MW wind turbine, for small perturbations about a mean operating state.

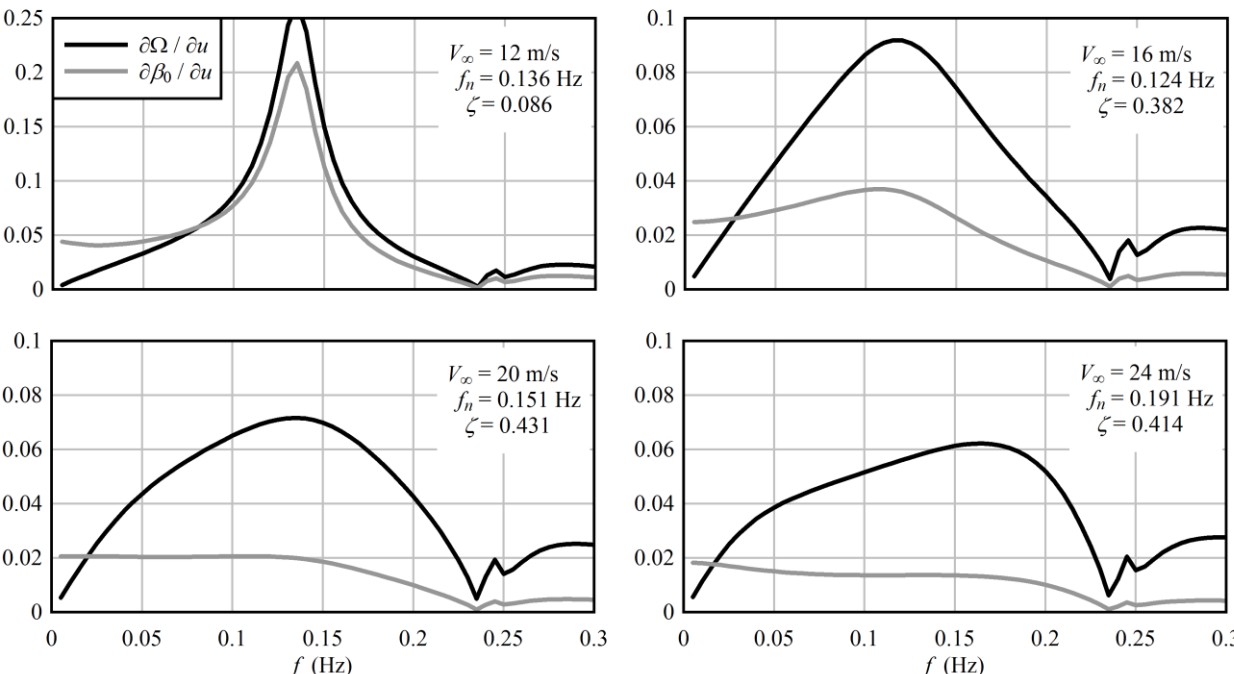

Figure 2: Closed-loop transfer functions of collective blade pitch (gray lines) and rotor speed (black lines) with respect to a uniform sinusoidal perturbation in the axial windspeed. Angular units are radians. Note the different Y-axis scale of the upper-left plot.



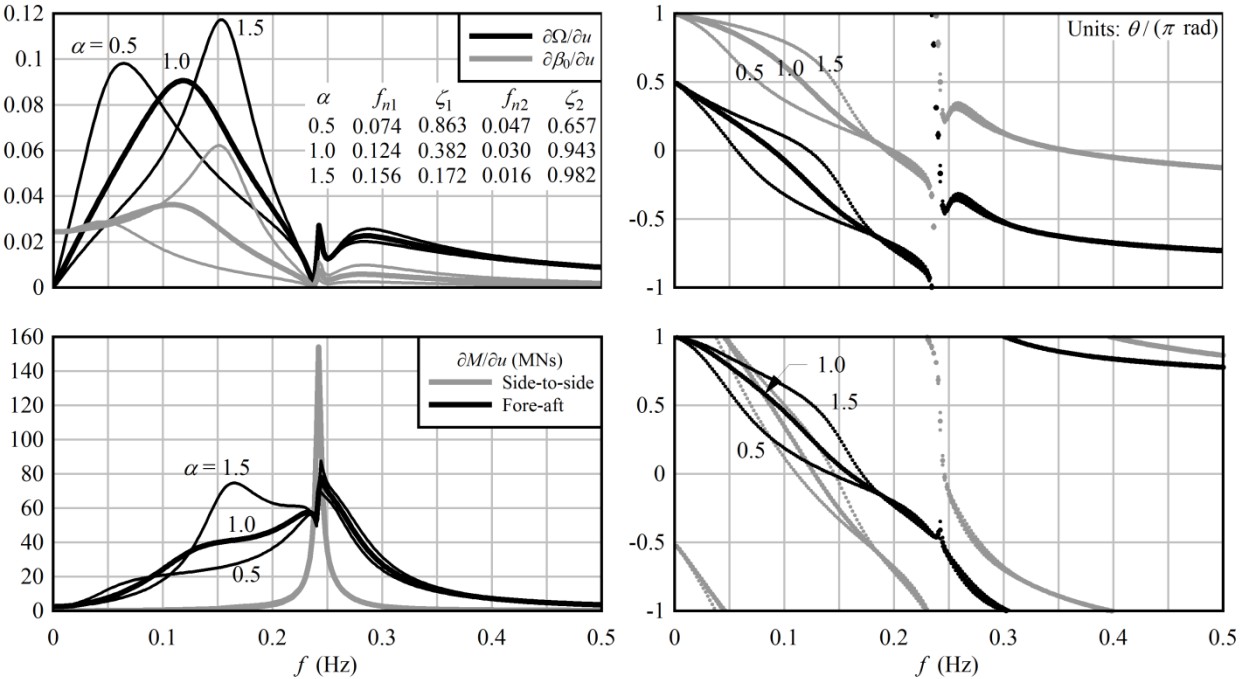

**Figure 3: Magnitudes (left) and phases (right) of transfer functions between axial windspeed and blade pitch, rotor speed, and tower mudline bending moments. Three gains are shown: 0.5, 1.0 (thick lines), and 1.5 times the baseline gains from Fig. 1. The natural frequency (Hz) and damping ratio of the rotor speed control (1) and dynamic wake (2) modes are also shown, tabulated as a function of the gain multiple.**

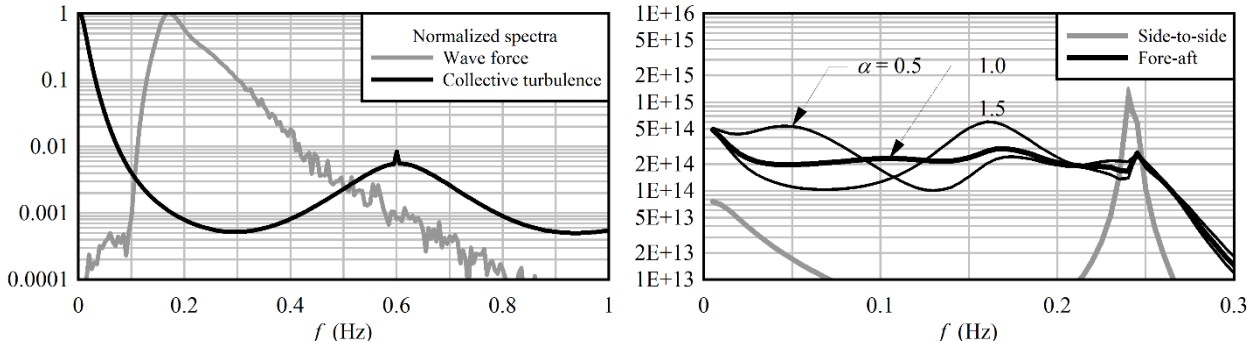

**Figure 4: At left: normalized spectra of the collective component of rotationally-sampled axial turbulence ($V_\infty = 16$ m/s, $I = 0.15$) at $r/R = 0.75$, and ocean wave loads with $H_s = 2$ m and $T_p = 6$ s. At right: spectra of tower bending moments at the mudline, for gain multiples of 0.5, 1.0 (thick line), and 1.5. (The three curves associated with the side-to-side spectra overlap.)**





**Figure 5: Closed-loop transfer functions of collective blade pitch (thin lines) and rotor speed (thick lines) with respect to a uniform sinusoidal perturbation in the axial windspeed. Nonlinear time-domain computations using FAST v8 are compared against equivalent results from a linear state-space model, obtained using equilibrium and dynamic wakes. The FAST results should be compared against the equilibrium-wake (black) curves.**



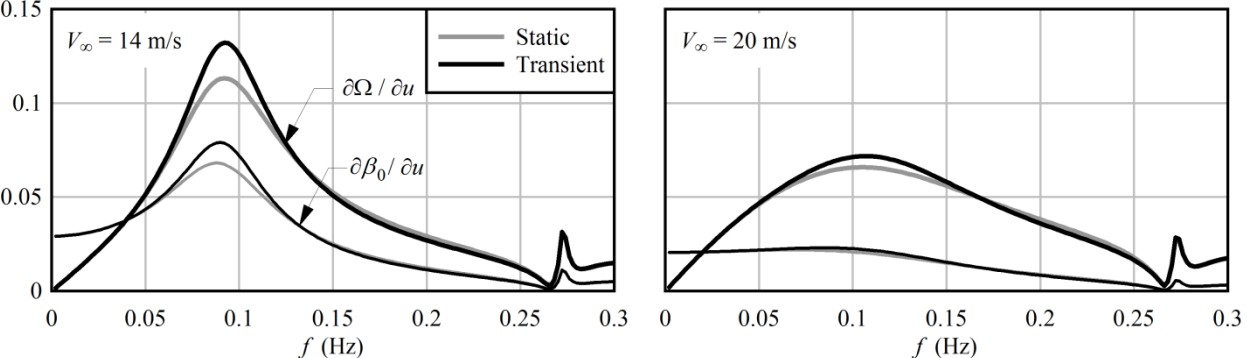

**Figure 6: The influence of transient circulation on the closed-loop transfer functions of collective blade pitch and rotor speed with respect to windspeed.**

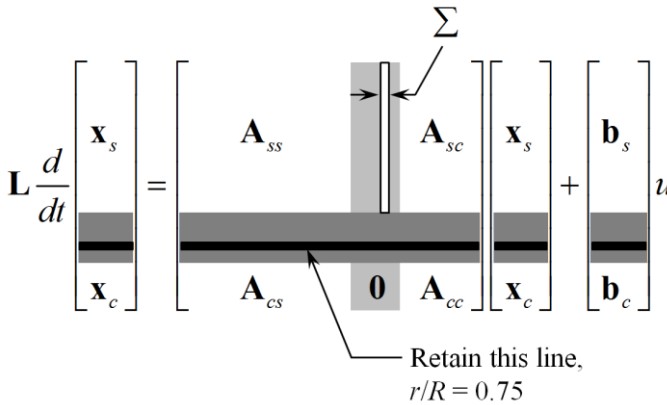

**Figure 7: A sketch of the matrix operations used to reduce the number of aerodynamic states. The input $u$ is the weighted rotor-average rotationally-sampled windspeed, by way of Eq. (13).**

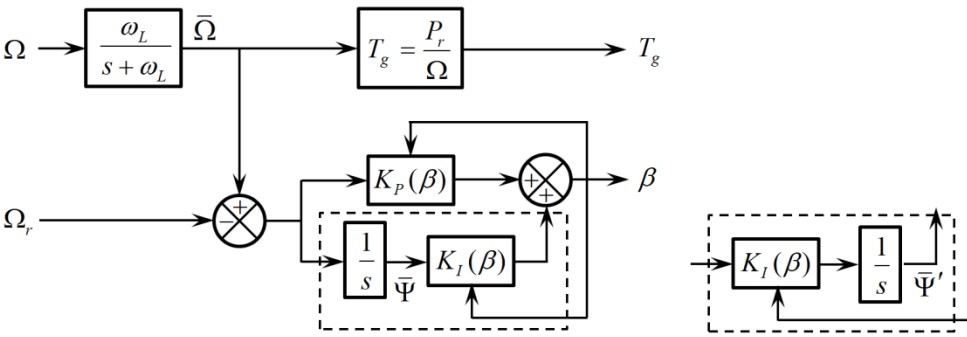

10  **Figure 8: At left, the equivalent functions of the NREL 5 MW turbine controller during normal, non-saturated operation. The integral pathway (dashed box) contains the scheduled gain outside the integrated speed error. At right, an alternate integral pathway with the scheduled gain inside the integral of the speed error.**




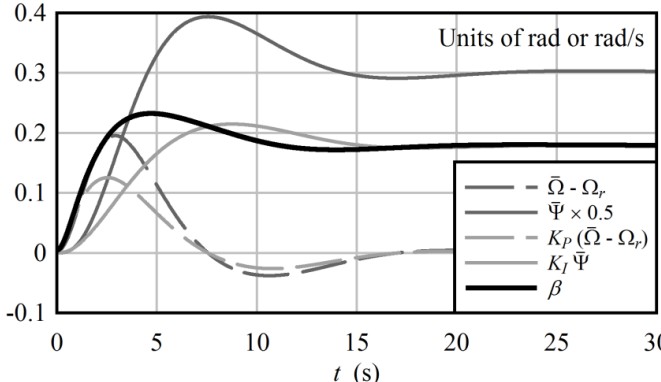

**Figure 9: Startup of a simulation at a windspeed of 15 m/s, showing how the steady-state blade pitch angle is set by the integral pathway. The perceptible lag between the integrated speed error, and the integral pathway's contribution to the blade pitch command, hints at the problem with scheduling the gain outside the integral of the speed error.**

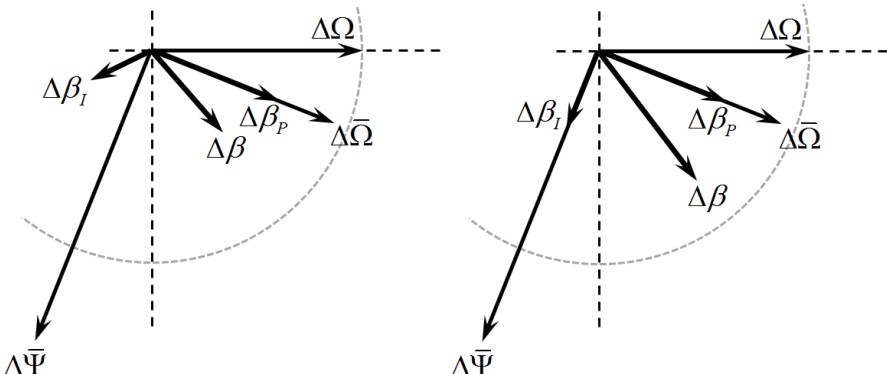

**Figure 10: A phasor diagram of the controller dynamics for gain scheduling (at left) outside the integral and (at right) inside the integral. The magnitudes and phases are normalized with respect to the shaft speed input; the dashed gray line indicates the unit circle.**



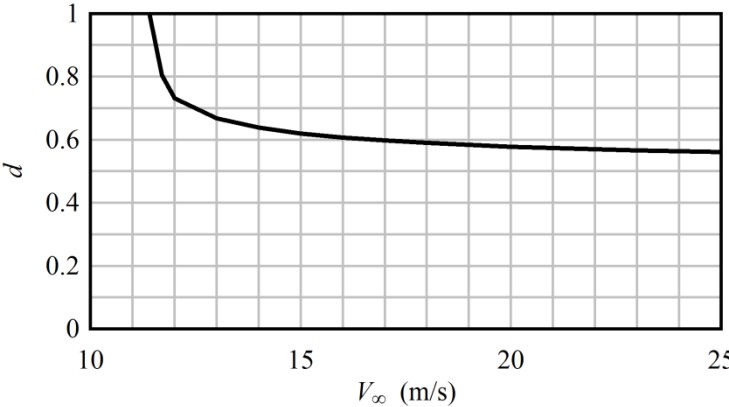

**Figure 11: The factor giving the effective gains of the NREL 5 MW controller, with respect to the nominal values, which have been scheduled outside the integral.**

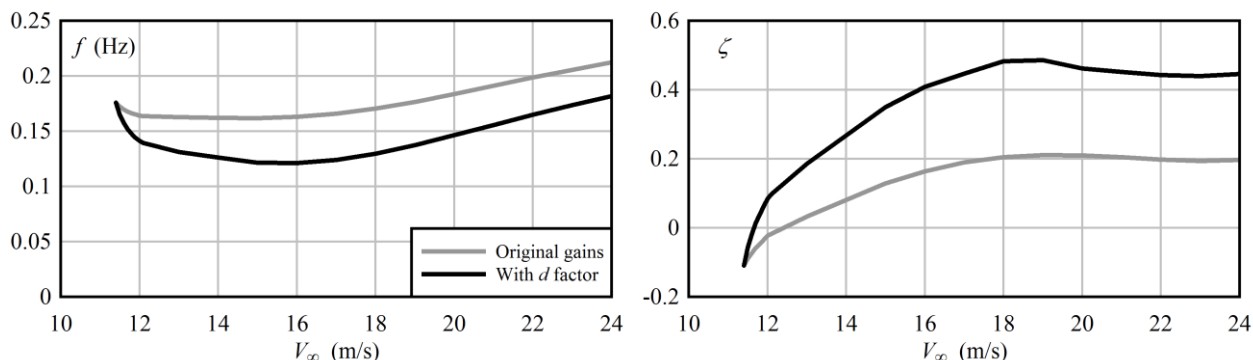

**Figure 12: The natural frequency and damping ratio of the rotor speed control mode, where the integral gain has been properly scheduled inside the integrator, comparing the performance of the original published gains with the case where the gains are reduced by the $d$ factor.**



(A): $\beta$ (deg), left axis.  (B): $10(V_\infty - 12)$ (m/s), left axis.  (C): $10(\Omega - \Omega_r)$ (rad/s), right axis.

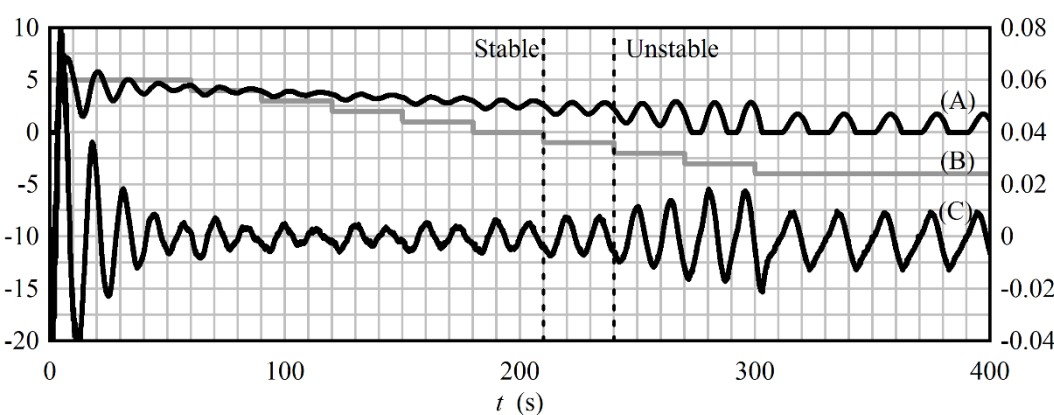

**Figure 13: A FAST v8/BeamDyn time-domain analysis of the NREL 5 MW turbine with baseline controller, showing unstable behavior at windspeeds below 11.9 m/s.  The amplitude of the unstable behavior is bounded by nonlinearities: the control mode transition on one side, and a stable region, existing because of gain scheduling, on the other.**

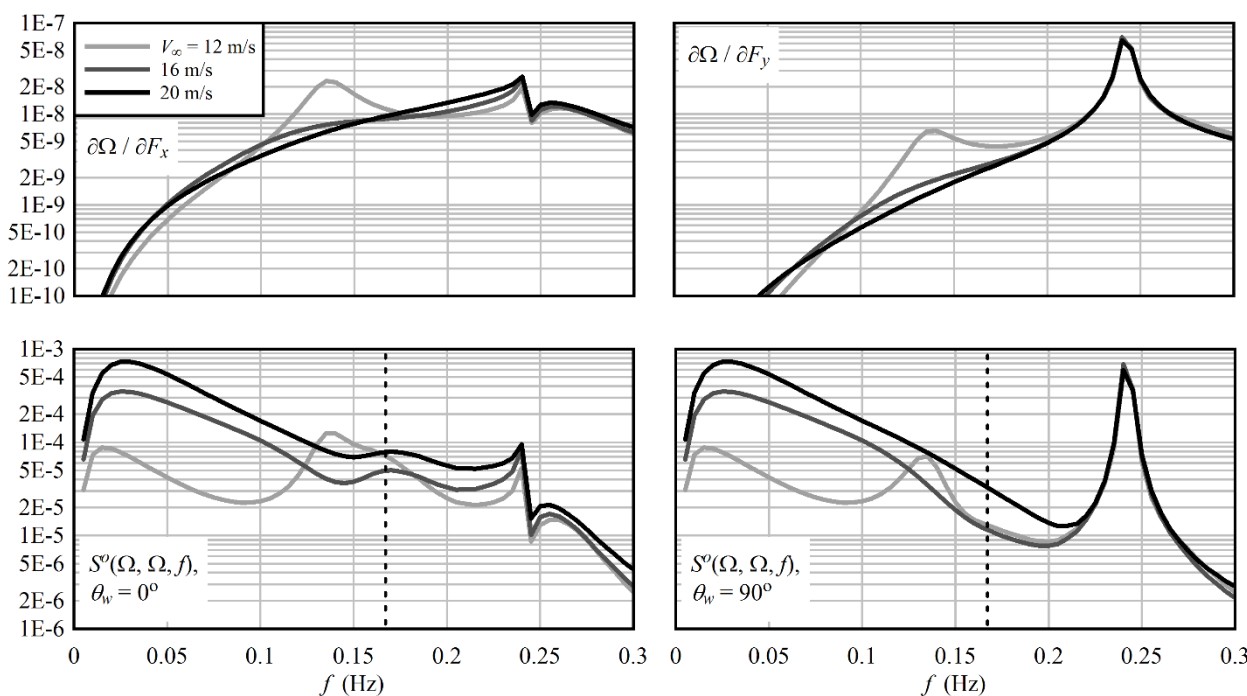

**Figure 14: Transfer functions (above) of waterline wave forces to rotor speed, and spectra (below) for a wave state of $H_s = 2$ m, $T_p = 6$ s.  At left: wind and waves aligned.  At right: waves orthogonal to wind.  The dashed line indicates the peak in the wave elevation spectrum.**





**Figure 15: Spectra of rotor speed, nacelle fore-aft displacement, and blade pitch acceleration, in small-amplitude ($I = 0.02$) turbulence. The simplified Model 7D, with only one rotor-average windspeed input, is compared against a full (572-state) linear model and nonlinear time-domain simulations. At left: mean windspeed of 16 m/s; at right: 20 m/s. The $x$ axis of the pitch acceleration plots is extended in order to include 3P rotationally-sampled turbulence.**





**Figure 16: The results of Figure 16, repeated with $I$ = 0.16. At left: mean windspeed of 16 m/s; at right: 20 m/s. The figure at lower left is the spectrum of blade pitch angle; the spectrum of pitch acceleration is meaningless, due to the abrupt saturation of the pitch angle at 0°.**



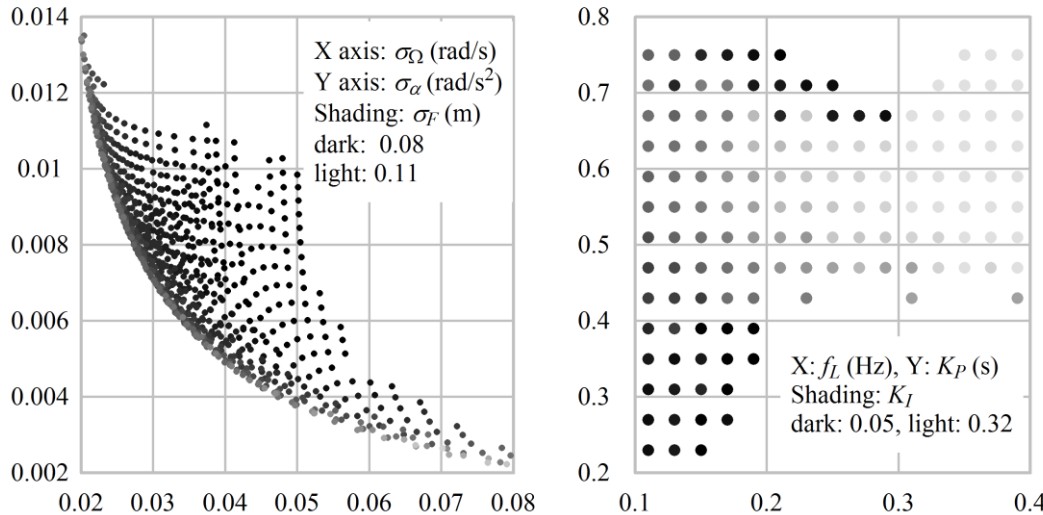

**Figure 17: Points on the Pareto front, plotted according to the objectives (at left) and the control tuning parameters (at right). The windspeed is 16 m/s.**

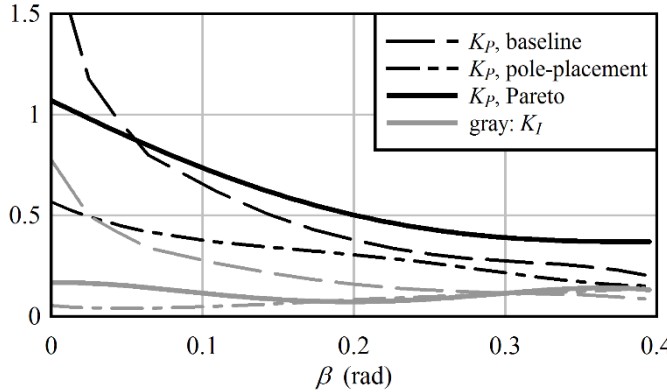

**Figure 18: A comparison of the gains, scheduled as a function of the pitch angle.**





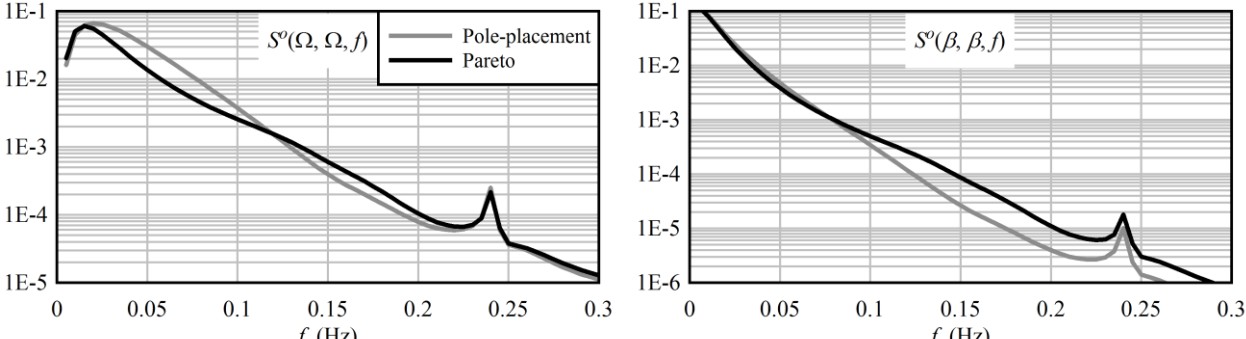

**Figure 19: An example of how tighter speed control is obtained by increasing the frequency of the rotor speed control mode. The windspeed is 16 m/s.**



**Table I: Natural frequencies and damping ratios of the rotor speed control mode (using the baseline gains) and dynamic wake mode (at half the baseline gains), obtained from models with various degrees-of-freedom. Results obtained with the reference high-fidelity model are highlighted in bold. Model 5D is recommended as a minimum model for basic controller gain tuning.**

| | | | | Rotor speed control mode, $\alpha = 1.0$ | | | | | |
| | | | | 12 m/s | | 16 m/s | | 20 m/s | |
| ID | DOFs | $N_s$ | Aero | $f$ | $\zeta$ | $f$ | $\zeta$ | $f$ | $\zeta$ |
|---|---|---|---|---|---|---|---|---|---|
| 1 | R | 3 | QS | 0.0579 | 0.3336 | 0.0578 | 0.5246 | 0.0474 | 0.7648 |
| 2 | R d | 5 | QS | 0.0580 | 0.3381 | 0.0578 | 0.5267 | 0.0472 | 0.7664 |
| 3e | R d e | 7 | QS | 0.0580 | 0.3385 | 0.0579 | 0.5240 | 0.0476 | 0.7638 |
| 3f[3] | R d f | 7 | QS | 0.0546 | 0.2068 | 0.0602 | 0.4575 | 0.0531 | 0.7283 |
| 3F[4] | F R d | 7 | QS | 0.0594 | 0.3152 | 0.0594 | 0.5157 | 0.0487 | 0.7680 |
| 4 | R d f t | 9 | QS | 0.0789 | 0.2293 | 0.0746 | 0.4922 | 0.0759 | 0.7717 |
| 5Q | F R d f t | 11 | QS | 0.0804 | 0.1889 | 0.0782 | 0.4621 | 0.0892 | 0.7101 |
| 5D | F R d f t | 16 | Dyn[1] | 0.1321 | 0.1700 | 0.1141 | 0.4753 | 0.1425 | 0.5203 |
| 6D | S F R d f e | 18 | Dyn[1] | 0.0824 | 0.4710 | 0.0656 | 0.5708 | 0.0874 | 0.8212 |
| 7Q[5] | S F R d f e t | 15 | QS | 0.0804 | 0.1902 | 0.0782 | 0.4569 | 0.0904 | 0.6939 |
| 7D | S F R d f e t | 20 | Dyn[1] | 0.1318 | 0.1722 | 0.1128 | 0.4697 | 0.1401 | 0.5124 |
| 8D | S F R d f e t2 | 22 | Dyn[1] | 0.1360 | 0.1338 | 0.1185 | 0.4248 | 0.1443 | 0.4757 |
| | Full | 572 | QS | 0.0889 | 0.1706 | 0.0847 | 0.4404 | 0.1040 | 0.6295 |
| | Full | 221 | Dyn[1] | 0.1385 | 0.1124 | 0.1215 | 0.4008 | 0.1464 | 0.4560 |
| | **Full** | **572** | **Dyn[2]** | **0.1370** | **0.1050** | **0.1211** | **0.4087** | **0.1465** | **0.4618** |
| | | | | Dynamic wake mode, $\alpha = 0.5$ | | | | | |
| 5D | F R d f t | 16 | Dyn[1] | 0.0604 | 0.8445 | 0.0464 | 0.6092 | 0.0172 | 0.9208 |
| 6D | S F R d f e | 18 | Dyn[1] | 0.0477 | 0.9365 | 0.0661 | 0.9297 | 0.000 | > 1 |
| 7D | S F R d f e t | 20 | Dyn[1] | 0.0602 | 0.8450 | 0.0466 | 0.6065 | 0.0171 | 0.9210 |
| 8D | S F R d f e t2 | 22 | Dyn[1] | 0.0582 | 0.8399 | 0.0479 | 0.6189 | 0.0162 | 0.9282 |
| | Full | 221 | Dyn[1] | 0.0563 | 0.8408 | 0.0487 | 0.6280 | 0.0156 | 0.9324 |
| | **Full** | **572** | **Dyn[2]** | **0.0521** | **0.8762** | **0.0455** | **0.6455** | **0.0161** | **0.9243** |

R: rigid rotor and blade pitch. d: driveshaft torsion. e: blade edgewise. f: blade flapwise. t: blade torsion. F: tower fore-aft. S: tower side-to-side. Notes: (1) Reduced rotor-average transient circulation, stall, and wake models: 5 states. (2) Full transient circulation, stall, and wake models using BEM. (3) The minimum elastic DOFs recommended by Wright. (4) The minimum elastic DOFs recommended by Bossanyi. (5) The minimum elastic DOFs recommended by Sønderby.

**Table II: Values of the standard deviation metrics, derived from Figures 15 and 16.**

| $I$ | $V_\infty$ | $\sigma_\Omega$ | | | $\sigma_\alpha$ | | | $\sigma_F$ | | |
| | | 7D | Lin. | FAST | 7D | Lin. | FAST | 7D | Lin. | FAST |
|---|---|---|---|---|---|---|---|---|---|---|
| 0.02 | 16 | 0.00556 | 0.00556 | 0.00610 | 0.00073 | 0.00081 | 0.00092 | 0.00842 | 0.00856 | 0.00754 |
| | 20 | 0.00727 | 0.00733 | 0.00793 | 0.00080 | 0.00091 | 0.00103 | 0.00879 | 0.00974 | 0.00805 |
| 0.16 | 16 | 0.0445 | 0.0445 | 0.0480 | 0.00590 | 0.00653 | n/a | 0.0674 | 0.0684 | 0.0708 |
| | 20 | 0.0582 | 0.0586 | 0.0600 | 0.00649 | 0.00736 | 0.00895 | 0.0703 | 0.0779 | 0.0669 |




**Table III: The pole-placement tuning and resulting metrics. The low-pass filter frequency $f_L$ is 0.20 Hz in all cases.**

| $V_\infty$ | $\beta_0$ | $K_P$ | $K_I$ | $\sigma_F$ | $\sigma_S$ | $\sigma_\Omega$ | $\sigma_\beta$ | $\sigma_\alpha$ | $\sigma_d$ | $\sigma_f$ | $\sigma_e$ |
|---|---|---|---|---|---|---|---|---|---|---|---|
| 12 | 0.064 | 0.420 | 0.040 | 0.143 | 0.052 | 0.0930 | 0.069 | 0.0030 | 0.0036 | 1.223 | 0.040 |
| 13 | 0.109 | 0.370 | 0.050 | 0.112 | 0.052 | 0.0726 | 0.053 | 0.0031 | 0.0024 | 0.921 | 0.023 |
| 14 | 0.143 | 0.350 | 0.066 | 0.101 | 0.052 | 0.0612 | 0.048 | 0.0034 | 0.0019 | 0.805 | 0.024 |
| 15 | 0.173 | 0.330 | 0.075 | 0.096 | 0.052 | 0.0574 | 0.045 | 0.0036 | 0.0017 | 0.738 | 0.029 |
| 16 | 0.201 | 0.310 | 0.085 | 0.094 | 0.051 | 0.0551 | 0.044 | 0.0039 | 0.0015 | 0.704 | 0.035 |
| 17 | 0.226 | 0.280 | 0.090 | 0.094 | 0.052 | 0.0559 | 0.043 | 0.0040 | 0.0014 | 0.687 | 0.041 |
| 18 | 0.250 | 0.260 | 0.095 | 0.094 | 0.052 | 0.0563 | 0.043 | 0.0041 | 0.0013 | 0.675 | 0.046 |
| 19 | 0.272 | 0.240 | 0.105 | 0.097 | 0.052 | 0.0563 | 0.043 | 0.0043 | 0.0012 | 0.671 | 0.052 |
| 20 | 0.294 | 0.220 | 0.107 | 0.099 | 0.053 | 0.0580 | 0.043 | 0.0044 | 0.0011 | 0.668 | 0.057 |
| 21 | 0.315 | 0.205 | 0.115 | 0.102 | 0.053 | 0.0584 | 0.043 | 0.0047 | 0.0011 | 0.668 | 0.063 |
| 22 | 0.335 | 0.188 | 0.120 | 0.105 | 0.054 | 0.0598 | 0.043 | 0.0048 | 0.0010 | 0.670 | 0.068 |
| 23 | 0.354 | 0.174 | 0.128 | 0.109 | 0.054 | 0.0605 | 0.044 | 0.0051 | 0.0010 | 0.674 | 0.073 |
| 24 | 0.374 | 0.160 | 0.134 | 0.113 | 0.055 | 0.0616 | 0.044 | 0.0053 | 0.0010 | 0.677 | 0.078 |
| 25 | 0.392 | 0.146 | 0.138 | 0.118 | 0.056 | 0.0632 | 0.044 | 0.0054 | 0.0010 | 0.681 | 0.083 |

5  **Table IV: The selected Pareto tuning and resulting metrics. The low-pass filter frequency $f_L$ is 0.17 Hz in all cases.**

| $V_\infty$ | $\beta_0$ | $K_P$ | $K_I$ | $\sigma_F$ | $\sigma_S$ | $\sigma_\Omega$ | $\sigma_\beta$ | $\sigma_\alpha$ | $\sigma_d$ | $\sigma_f$ | $\sigma_e$ |
|---|---|---|---|---|---|---|---|---|---|---|---|
| 12 | 0.064 | 0.846 | 0.144 | 0.135 | 0.047 | 0.0347 | 0.067 | 0.0057 | 0.0033 | 1.185 | 0.034 |
| 13 | 0.109 | 0.717 | 0.110 | 0.104 | 0.048 | 0.0361 | 0.052 | 0.0054 | 0.0022 | 0.865 | 0.018 |
| 14 | 0.143 | 0.621 | 0.088 | 0.093 | 0.048 | 0.0402 | 0.047 | 0.0052 | 0.0018 | 0.742 | 0.020 |
| 15 | 0.173 | 0.550 | 0.077 | 0.089 | 0.049 | 0.0438 | 0.044 | 0.0052 | 0.0016 | 0.676 | 0.025 |
| 16 | 0.201 | 0.499 | 0.074 | 0.087 | 0.049 | 0.0461 | 0.043 | 0.0052 | 0.0014 | 0.643 | 0.030 |
| 17 | 0.226 | 0.462 | 0.078 | 0.087 | 0.049 | 0.0468 | 0.042 | 0.0054 | 0.0013 | 0.624 | 0.035 |
| 18 | 0.250 | 0.435 | 0.087 | 0.089 | 0.049 | 0.0466 | 0.042 | 0.0057 | 0.0012 | 0.616 | 0.041 |
| 19 | 0.272 | 0.414 | 0.099 | 0.092 | 0.049 | 0.0461 | 0.042 | 0.0060 | 0.0011 | 0.613 | 0.046 |
| 20 | 0.294 | 0.398 | 0.112 | 0.095 | 0.050 | 0.0457 | 0.042 | 0.0065 | 0.0011 | 0.613 | 0.051 |
| 21 | 0.315 | 0.386 | 0.125 | 0.099 | 0.050 | 0.0454 | 0.042 | 0.0070 | 0.0010 | 0.615 | 0.056 |
| 22 | 0.335 | 0.376 | 0.136 | 0.104 | 0.051 | 0.0455 | 0.043 | 0.0075 | 0.0009 | 0.617 | 0.060 |
| 23 | 0.354 | 0.369 | 0.142 | 0.109 | 0.051 | 0.0461 | 0.043 | 0.0080 | 0.0009 | 0.618 | 0.065 |
| 24 | 0.374 | 0.368 | 0.142 | 0.114 | 0.052 | 0.0467 | 0.043 | 0.0086 | 0.0009 | 0.615 | 0.069 |
| 25 | 0.392 | 0.375 | 0.135 | 0.119 | 0.052 | 0.0477 | 0.042 | 0.0092 | 0.0009 | 0.610 | 0.072 |



**Table V: Modal frequency and damping properties of the pole-placement tuning.**

| $V_\infty$ | $f_{DW}$ | $\zeta_{DW}$ | $f_{RSC}$ | $\zeta_{RSC}$ | $f_S$ | $\zeta_S$ | $f_F$ | $\zeta_F$ | $f_f$ | $\zeta_f$ | $f_d$ | $\zeta_d$ |
|---|---|---|---|---|---|---|---|---|---|---|---|---|
| 12 | 0.008 | 0.905 | 0.101 | 0.617 | 0.242 | 0.0063 | 0.249 | 0.090 | 0.891 | 0.682 | 1.850 | 0.078 |
| 13 | 0.009 | 0.924 | 0.101 | 0.621 | 0.242 | 0.0062 | 0.249 | 0.093 | 0.912 | 0.676 | 1.849 | 0.077 |
| 14 | 0.009 | 0.953 | 0.101 | 0.617 | 0.242 | 0.0062 | 0.249 | 0.097 | 0.911 | 0.681 | 1.847 | 0.076 |
| 15 | 0.009 | 0.967 | 0.102 | 0.618 | 0.242 | 0.0063 | 0.248 | 0.100 | 0.903 | 0.693 | 1.844 | 0.075 |
| 16 | 0.008 | 0.984 | 0.102 | 0.609 | 0.242 | 0.0064 | 0.248 | 0.102 | 0.910 | 0.695 | 1.840 | 0.075 |
| 17 | 0.005 | 0.994 | 0.100 | 0.624 | 0.242 | 0.0065 | 0.247 | 0.104 | 0.917 | 0.696 | 1.837 | 0.074 |
| 18 | 0.004 | 0.996 | 0.102 | 0.620 | 0.242 | 0.0066 | 0.246 | 0.107 | 0.923 | 0.697 | 1.833 | 0.073 |
| 19 | 0.000 | >1 | 0.100 | 0.619 | 0.242 | 0.0068 | 0.246 | 0.109 | 0.929 | 0.699 | 1.829 | 0.072 |
| 20 | 0.000 | >1 | 0.101 | 0.628 | 0.242 | 0.0070 | 0.245 | 0.112 | 0.934 | 0.702 | 1.824 | 0.071 |
| 21 | 0.000 | >1 | 0.102 | 0.620 | 0.242 | 0.0071 | 0.244 | 0.114 | 0.939 | 0.704 | 1.819 | 0.070 |
| 22 | 0.000 | >1 | 0.101 | 0.627 | 0.242 | 0.0074 | 0.243 | 0.116 | 0.945 | 0.707 | 1.815 | 0.069 |
| 23 | 0.000 | >1 | 0.101 | 0.618 | 0.242 | 0.0076 | 0.242 | 0.118 | 0.952 | 0.711 | 1.810 | 0.069 |
| 24 | 0.035 | 0.984 | 0.101 | 0.619 | 0.242 | 0.0078 | 0.241 | 0.119 | 0.975 | 0.725 | 1.804 | 0.068 |
| 25 | 0.061 | 0.954 | 0.100 | 0.623 | 0.242 | 0.0080 | 0.240 | 0.120 | 0.962 | 0.712 | 1.799 | 0.067 |

5   **Table VI: Modal frequency and damping properties of the selected Pareto tuning.**

| $V_\infty$ | $f_{DW}$ | $\zeta_{DW}$ | $f_{RSC}$ | $\zeta_{RSC}$ | $f_S$ | $\zeta_S$ | $f_F$ | $\zeta_F$ | $f_f$ | $\zeta_f$ | $f_d$ | $\zeta_d$ |
|---|---|---|---|---|---|---|---|---|---|---|---|---|
| 12 | 0.000 | >1 | 0.129 | 0.242 | 0.242 | 0.0064 | 0.254 | 0.101 | 0.901 | 0.680 | 1.836 | 0.077 |
| 13 | 0.000 | >1 | 0.133 | 0.268 | 0.242 | 0.0064 | 0.254 | 0.105 | 0.919 | 0.674 | 1.840 | 0.077 |
| 14 | 0.000 | >1 | 0.135 | 0.307 | 0.242 | 0.0064 | 0.252 | 0.108 | 0.916 | 0.680 | 1.841 | 0.076 |
| 15 | 0.004 | 0.985 | 0.136 | 0.340 | 0.242 | 0.0065 | 0.251 | 0.111 | 0.907 | 0.692 | 1.840 | 0.075 |
| 16 | 0.003 | 0.993 | 0.139 | 0.356 | 0.242 | 0.0066 | 0.250 | 0.114 | 0.913 | 0.694 | 1.838 | 0.074 |
| 17 | 0.003 | 0.993 | 0.142 | 0.361 | 0.242 | 0.0068 | 0.249 | 0.117 | 0.919 | 0.695 | 1.835 | 0.074 |
| 18 | 0.005 | 0.989 | 0.145 | 0.359 | 0.242 | 0.0069 | 0.248 | 0.122 | 0.926 | 0.697 | 1.831 | 0.073 |
| 19 | 0.006 | 0.984 | 0.149 | 0.353 | 0.242 | 0.0071 | 0.247 | 0.127 | 0.931 | 0.699 | 1.827 | 0.072 |
| 20 | 0.008 | 0.982 | 0.153 | 0.343 | 0.242 | 0.0073 | 0.246 | 0.132 | 0.937 | 0.701 | 1.823 | 0.071 |
| 21 | 0.009 | 0.982 | 0.158 | 0.331 | 0.242 | 0.0075 | 0.245 | 0.139 | 0.943 | 0.703 | 1.818 | 0.070 |
| 22 | 0.009 | 0.983 | 0.164 | 0.319 | 0.242 | 0.0077 | 0.244 | 0.147 | 0.949 | 0.706 | 1.813 | 0.069 |
| 23 | 0.010 | 0.999 | 0.172 | 0.307 | 0.242 | 0.0080 | 0.242 | 0.158 | 0.956 | 0.710 | 1.808 | 0.069 |
| 24 | 0.024 | 0.995 | 0.184 | 0.288 | 0.242 | 0.0083 | 0.239 | 0.176 | 0.978 | 0.723 | 1.803 | 0.068 |
| 25 | 0.030 | 0.993 | 0.200 | 0.245 | 0.242 | 0.0087 | 0.233 | 0.210 | 0.966 | 0.711 | 1.798 | 0.067 |