# Peer review of "Basic controller tuning for large offshore wind turbines"

_Wind Energy Science, 2016_

## Referee Comment (RC1) · Anonymous Referee #1 · 7 Jun 2016

General comments:

This paper would serve well as a tutorial. Explanations are good, and many relevant issues are covered. However it does not really advance the field: I struggle to find any new insights or ideas. There is a narrow focus on one specific turbine example, which happens to come with a very rudimentary and inadequate controller. It is unfortunate that this one example has been used so extensively in academic circles, often without questioning the basis of its controller. The paper would be greatly improved by just a few comments recognising the limitations of this particular case study. Other researchers have designed their own controllers for this turbine, and therefore have not necessarily encountered the problems which this paper dwells heavily on; and outside of academia there are very many commercial turbine designs with good, professionally-designed controllers.

Specific comments:

[Figure]

Model order: There is quite some emphasis on which degrees of freedom should be used, but not enough recognition that this actually depends on the specific case. When designing a controller, it is always important to analyse and understand the dynamics of the plant so that the appropriate degrees of freedom are used. A turbine with a stiff tower, for example, is very different from one with a soft tower. In general one can start with all practical degrees of freedom and then reduce them to what's needed, rather than trying to decide in advance. There are methods for this based on controllability and observability. Such formal model order reduction methods can also be used for the aerodynamic states, rather than arbitrarily picking a blade element at 75% span.

Controller structure: the problem of gain scheduling which this paper dwells on is nothing special: it is specific to the particular controller structure which was chosen. One always has to consider the controller gains in the context of the chosen controller structure. A classical PI-based wind turbine controller need not be like this example, but can also include all sorts of loop-shaping filters, auxiliary loops and other devices to tune the response. The paper should acknowledge this, and also the key goal of reducing fatigue loading which is not really captured by the simple performance metrics used.

Controller tuning and sensitivity: part of a serious controller design is to make sure it is robust, and will remain stable in all sorts of environmental conditions, not all of which can be simulated, as well as to modelling imperfections, as no simulation model can be trusted to model the system perfectly - and it's not just the model structure, but also the parameters, whose values are likely to be different in the real as-built turbine. Such robustness was clearly not considered in designing the original controller, which has probably been used far beyond its original intention. There are classical control design techniques which help to ensure robustness, reducing the chance that simply changing to a different simulation model results in instability. That would clearly be unsatisfactory, as there is no guarantee that it would be stable on the real turbine.

---

## Referee Comment (RC2) · Anonymous Referee #2 · 28 Jul 2016

General comment

The paper presents a set of interesting investigations for the NREL 5 MW reference wind turbine. The paper is very well written. Complex issues are explained in a good way. However, there are three main issues, which weaken the paper:

1. The paper addresses several issues and thus tends to be a conglomeration of investigations. The investigation about the impact of the model fidelity on the closed-loop transfer function (Section 2) seems to be the most interesting, while Section 3-6 are less useful (see detailed comments below).

2. The paper focuses on the 5 MW reference wind turbine and then extrapolates the findings to (large offshore) wind turbines.

3. The most important finding is partly trivial ("the appropriate controller tuning is highly dependent on the aeroelastic model") and partly not correct ("therefore, no single reference controller can be defined, for use with all models").

a. One major task (if not the most important one) in model based controller design (most of modern control) is finding the appropriate controller design model. Thus it is trivial, that the control tuning is dependent on the model.

b. The question, if a controller can handle the uncertainties / simplification is addressed by its robustness (robust stability / performance). Thus, if a controller is robust enough, a single controller can be defined for use with all models considered in the robust design. The example of the instable controller is not a proof that no controller can handle all possible model fidelities of wind turbines.

Recommendation

The paper could be improved by

1. Focusing on the impact of the model fidelity on the closed-loop transfer function. Reducing / removing Section 3-6. The title should then be adjusted.

2. Stating more clearly that the findings are specific for the used reference wind turbine and explain the limitations of the findings.

3. Reformulating the most important finding.

Details Section 3 (Gain Scheduling)

Multiplying the gain after the integration and thus multiplying it to a non-zero-mean signal is an implementation error, since the controller is designed by shaping the linear closed-loop. Dedicating almost a full section to this issue seems to be exaggerated. Further, it can be assumed that most researcher have - knowingly or unknowingly - implemented the controller correctly. The instability issue when using a model with blade-torsion is interesting. However, to call into question other papers using the reference controller (with or without the implementation error) based on this investigation is very questionable.

Details Section 4 (wave-driven tower resonance)

Usually, notch filters for pitch and torque are used to avoid resonance (besides of the mentioned active damping). The recommendation to lower the low-pass filter cut-off frequency to a value well below the first natural frequency of the tower seems to be not very helpful, because the performance of the pitch and torque control loop should be reduced due to the increasing delay.

Details Section 5 and 6

Controller tuning based on multiplication of disturbance spectra and closed loop transfer function is smart. However, it is not really new and has been proposed before. Further, important measures such as DEL are not considered. Then, it is not clear, how the polynomial of 4th order has been obtained and why this order has been chosen (a simple interpolation might be more straight-forward to implement).

Less important issues

- It is not clear, if the spectra from Fig. 4 is obtained from a time domain simulation or by multiplication of the wind/wave spectra? Sorry, if I missed it.

- Why different linear models are used in Fig. 3 and 5 / Section 2.1 and 2.2? Is it only because of the stiff seabed in FAST? If so, might be helpful to mention it shortly, e.g. by "…0.24 Hz, see Fig. 3 and Section 2.1; the difference…"

- Units are sometimes missing, e.g. in eq.(14)+(15).

- Please consider to use vector-based graphs which might increase readability of figures.

- The Nomenclature is nice, but explaining the used symbols closer to the equations might increase the readability of the paper.

---

## Author Comment (AC1) · 28 Jul 2016

I thank the reviewers for their critical comments. At this stage of the review process, I have been asked to prepare a response, without revising the manuscript. Therefore, in lieu of making changes to the manuscript itself, I have summarized how I propose to resolve each comment.

————————————————————————————-

(A)

R1, general comment: [The paper] does not really advance the field: I struggle to find any new insights or ideas. There is a narrow focus on one specific turbine example, which happens to come with a very rudimentary and inadequate controller. It is unfortunate that this one example has been used so extensively in academic circles, often without questioning the basis of its controller.

R2, general comments 2 and 3: The paper focuses on the 5 MW reference wind turbine and then extrapolates the findings to (large offshore) wind turbines... The most important finding is partly trivial ("the appropriate controller tuning is highly dependent on the aeroelastic model") and partly not correct ("therefore, no single reference controller can be defined, for use with all models").

—————-

Working in the field of wind turbine control is challenging, as the controls influence the dynamics of the entire system. Few aspects of wind turbine or wind power plant dynamics can be studied without considering the controls. As a consequence, the wind turbine controller is not the province of a select group of controls experts; just about everyone in the wind energy research and engineering community has a stake.

If the principle findings of the paper are well-known within a subset of researchers on wind turbine controls – and it is indeed a subset, as evinced by the large number of controls researchers who have used the NREL 5 MW discon.dll reference controller uncritically – then they have been remarkably silent on the topic. To my knowledge there is no thorough critical analysis of the NREL controller tuning in the literature. Historical publications on basic PI control design and tuning need to be revisited in the context of large offshore wind turbines. Until Dunne's manuscript (which I learned about after independently developing the analysis of Section 3) the gain scheduling implementation of the NREL controller had not been criticized. No one has identified that this controller is unstable with a "proper" aeroelastic model; not even the researchers at NREL were aware that this is the case.

We must be careful in defining "the field" to which this paper makes an important contribution. The contribution to control theory is indeed minor: the suggestion that a Pareto front approach be applied for gain tuning where a reliable cost function is not available. But the target audience is not control theoreticians, and I am not attempting to publish in a journal on control theory. The topic and findings should be highly relevant for large

swaths of the wind energy research community.

Elevating knowledge from a small, closed group of experts – which in this case has been comparatively silent on a relevant topic – to the research community overall is an important purpose of the application-oriented scientific literature such as Wind Energy Science, and valid grounds for publication of a scientific article.

The manuscript concludes that the appropriate controller tuning is highly dependent on the aeroelastic model; therefore, no single reference controller can be defined, for use with all models. In my opinion this is neither trivial nor incorrect, though perhaps the scope of the statement needs to be more precise. Much of wind energy research is based on reference wind turbines, including descriptions of the aerodynamics, structures, and controls. These reference turbines are implemented in a variety of models, from high-resolution 3D geometry for CFD/FEM, to models containing just a few degrees-of-freedom for electrical grid analysis. If the sensitive interdependence of the aeroelastic model and control tuning is trivial, why have so many publications from the low- or high-resolution ends of the spectrum not questioned the definition of the NREL 5 MW controller for their applications? The aeroelastic analysts with which I have shared the manuscript have been surprised at the degree to which the closed-loop response is sensitive to the choice of flexible degrees-of-freedom. It is vital for the next round of reference turbine definitions (for which I am coordinating the IEA Task) that we define the controller in terms of modal performance specifications, and not a single fixed tuning. This is likely applicable to MIMO state-space control designs as well; with a fixed control tuning here it would be the pole placement, or the effective cost function weights, which would be in disagreement between models of different resolutions. The only exception I can think of, where a single controller could give equivalent effective performance on all models, would be some sort of online adaptive controller; and this is a fringe case.

The NREL 5 MW wind turbine, being the established reference case representing large offshore wind turbines, is used as an example throughout the paper. I have also repeated the analysis of Table I on a 10 MW reference offshore wind turbine, with similar results as to which DOFs are required. The 10 MW wind turbine is based on the DTU 10 MW rotor and controller, with a nacelle, tower, and foundation designed by NTNU. The 10 MW design has not yet been published in full. To make the results reproduceable, I'd have to describe the turbine in the manuscript, which would lead to an unacceptably broad scope.

A wind turbine similar to this 10 MW design will be used as the next generation reference offshore wind turbine, released under the umbrella of the IEA Task. So the results of the paper, it may be stated, are applicable to "the set of reference wind turbines which will be the basis for much of the research on offshore wind energy over the next few years."

——————-

Proposed resolution:

A revised manuscript should be published, in order to bring attention within the wind energy community to the issues associated with basic controller design and implementation in dynamic simulations.

Clarify the context of the principal conclusion on the interdependence of the aeroelastic model and control tuning.

Mention that the results have been repeated on a 10 MW turbine, and that the results are limited to this class of offshore wind turbines.

———————————————————————-

(B)

R1, model order: In general one can start with all practical degrees of freedom and then reduce them to what's needed, rather than trying to decide in advance. There are methods for this based on controllability and observability.

—————

Sure, I could couch the analysis of Table I in the language and theorems of controllability and observability. The outcome would be the same, and it would be less comprehensible to the aeroelastic analyst.

—————

Proposed resolution:

State that equivalent results could be obtained by formal model reduction methods, also by observing which DOFs participate in the rotor speed control mode computed for the full model.

———————————————————————————-

(C)

R1, model order: Formal model order reduction methods can also be used for the aerodynamic states, rather than arbitrarily picking a blade element at 75% span.

—————

There is very little further improvement to be had, in the context of control tuning, as can be seen in Figure 15. Picking a representative element for the dynamic wake time response is nice because it is simple. Why add complexity, if it doesn't buy you anything?

—————

Proposed resolution:

Repeat the aerodynamic mode reduction with an established technique, and explain why picking a blade element at 75% span gives good results.

———————————————————————————-

(D)

R1, controller structure: The paper should acknowledge ... the key goal of reducing fatigue loading which is not really captured by the simple performance metrics used.

R2, Sections 5 and 6: Important measures such as DEL are not considered.
* * *
It is straightforward to go from an internal load spectrum to a good estimate of the DEL, by way of Dirlik's cycle counting method. I chose to avoid DELs, because of the complication of having to pick an exponent. Do I pick fiberglass, for the blades? Or steel, for the tower? I expect that a reasonable control tuning obtained by the selected standard deviation metrics will also be a reasonable control tuning from the perspective of DELs.
* * *
Proposed resolution:

Repeat the analysis of Section 6 with DEL metrics instead of standard deviations; update the results if they are notably different. If not, then comment that using DELs produced similar results.
* * *
(E)

R1, general comments: The paper would be greatly improved by just a few comments recognising the limitations of this particular case study. Other researchers have designed their own controllers for this turbine, and therefore have not necessarily encountered the problems which this paper dwells heavily on; and outside of academia there are very many commercial turbine designs with good, professionally-designed controllers.

R1, controller structure: A classical PI-based wind turbine controller need not be like this example, but can also include all sorts of loop-shaping filters, auxiliary loops and other devices to tune the response. The paper should acknowledge this.
* * *
The recent DTU Wind Energy Controller (Hansen and Henriksen, DTU Wind Energy E-0028, 2013) for the 10 MW turbine includes some additional functionality related in particular to the transition between control modes. When linearized, the above-rated controller is essentially identical in structure to the NREL 5 MW controller, whose architecture and tuning are in turn based on earlier DTU publications. The basic PI architecture continues to be relevant.
* * *
Proposed resolution:

Add text (similar to the closing statement in Section 2.1) emphasizing that the results are valid for the basic PI architecture, and that auxiliary loops were not considered.
* * *
(F)

R1, controller tuning and sensitivity: There are classical control design techniques which help to ensure robustness, reducing the chance that simply changing to a different simulation model results in instability. That would clearly be unsatisfactory, as there is no guarantee that it would be stable on the real turbine.

R2, general comment 3b: The question, if a controller can handle the uncertainties / simplification is addressed by its robustness (robust stability / performance). Thus, if a controller is robust enough, a single controller can be defined for use with all models considered in the robust design. The example of the instable controller is not a proof that no controller can handle all possible model fidelities of wind turbines.

——————

It seems to me that certain portions of the wind energy research community (including myself, until recently) have underappreciated just how flexible large wind turbines are. That is, changing to a different simulation model isn't necessarily that "simple", in terms of the impact on the dynamic response!

A robust controller may remain stable with a variety of models, but unless it is adaptive, or retuned for each model, the closed-loop dynamic reponse will not be the same.

——————

Proposed resolution:

Add a discussion of gain and phase margins of the recommended control tunings.

Clarify the context: that the definition of a wind turbine and its controller should give a similar response, in terms of the key output quantities, for a simple model used in an electrical grid analysis; an aeroelastic loads model; and a high-fidelity CFD simulation. Otherwise, we aren't looking at the same wind turbine! Part of the solution is to be aware of the control tuning issues addressed in the present manuscript.

————————————————————————-

(G)

R2, general comment 1: The paper addresses several issues and thus tends to be a conglomeration of investigations. The investigation about the impact of the model fidelity on the closed loop transfer function (Section 2) seems to be the most interesting, while Section 3-6 are less useful.

——————

Here are the principle findings of Sections 3-6, together with comments as to why they are significant:

(1) Gains should be scheduled inside the integral. If this is well-known, then why has no one – including many control researchers – published this criticism of the ubiquitous NREL 5 MW controller, which schedules the gains outside the integral?

(2) The NREL 5 MW controller, when used with a "proper" aeroelastic model, is unstable in a narrow operating region above the rated windspeed. This leads to the questioning of some of the results – say, comparisons in terms of DELs – which have been published, which have used the NREL 5 MW controller as a baseline case. The wind energy community needs to be aware of this, so that further anomalous results are not obtained! This is especially so now that FAST V8 is released with both the defective controller and a fully-flexible model which includes blade torsion.

(3) Ocean-wave driven tower resonance may appear in the control response of a direct-drive turbine, and the filter frequency should be chosen to suppress this.

(4) The averaging of turbulence spectra according to Equation (13), together with the reduced model of Section 2.3, accurately reproduce the low-frequency aeroelastic response of the wind turbine to atmospheric turbulence. This has applications in control tuning and rapid evaluation of wind turbine loads.

(5) Figure 16 provides evidence that linear models can closely reproduce the results of nonlinear simulations for the dynamic response of the wind turbine under normal operating conditions. This observation, which also builds on the results of (Merz 2015a), is of interest, because it hints that the closed-loop aeroelastic response can be computed using extremely rapid and numerically smooth frequency-domain methods. This has applications for control tuning, as shown in the manuscript.

(6) A Pareto front method may be used for control tuning, in cases where a reliable cost function is not known. I am not aware of previous applications of Pareto analysis for wind turbine control tuning. Figure 17 illustrates nicely the fundamental tradeoff between speed control and pitch activity.

WESD

(7) An updated tuning is recommended for the NREL 5 MW turbine controller. Sure, retuning a PI controller is by itself not a significant scientific achievement, but why should I only criticize the existing control tuning, without offering an alternative solution?
* * *
Proposed resolution:

Retain Sections 3-6 of the manuscript in revised form.

_________________________________________-

(H)

R2, Section 3: Multiplying the gain after the integration and thus multiplying it to a non-zero-mean signal is an implementation error, since the controller is designed by shaping the linear closed-loop. Dedicating almost a full section to this issue seems to be exaggerated. Further, it can be assumed that most researcher have - knowingly or unknowingly - implemented the controller correctly. The instability issue when using a model with blade-torsion is interesting. However, to call into question other papers using the reference controller (with or without the implementation error) based on this investigation is very questionable.
* * *
From above: If this is well-known, then why has no one – including many control researchers – published this criticism of the ubiquitous NREL 5 MW controller, which schedules the gains outside the integral?

I do not believe that most implementations of the NREL 5 MW controller are correct. The research groups I am aware of have all used the discon.dll file available from the NREL website – the implementation used in all the IEA OCx analyses – which schedules the gain outside the integral term.

The results of the present manuscript show that the NREL 5 MW controller is unstable

when used with a proper aeroelastic model. This controller has been used uncritically as a baseline in many international studies and publications on wind turbine control. Why does the reviewer feel that we should NOT be skeptical of the results obtained with the potentially unstable control tuning?

\_\_\_\_\_\_\_\_\_\_

Proposed resolution:

Retain Section 3 in revised form.

\_\_\_\_\_\_\_\_\_\_\_\_\_\_\_\_\_\_\_\_\_\_\_\_\_\_\_\_\_\_\_\_\_\_\_\_\_\_\_\_

(I)

R2, Section 4: Usually, notch filters for pitch and torque are used to avoid resonance (besides of the mentioned active damping). The recommendation to lower the low-pass filter cut-off frequency to a value well below the first natural frequency of the tower seems to be not very helpful, because the performance of the pitch and torque control loop should be reduced due to the increasing delay.

\_\_\_\_\_\_\_\_\_\_

Of what use would the high-pass portion of a notch-filtered signal be, in the context of the basic above-rated PI pitch control action? One might try to use a higher-order low-pass filter to try to get some control response at frequencies closer to the tower resonance frequency, but I do not think this is critical. In my view the proposed filter frequency of 0.17 Hz is a sound choice which, together with the proposed gain scheduling, results in good performance.

\_\_\_\_\_\_\_\_\_\_

Proposed resolution:

Retain the original control architecture, with first-order filter, which then leads to the

suggestion of a 0.17 Hz filter frequency. (This also works on the 10 MW turbine, whose first tower frequencies are similar to the 5 MW.)

————————————————————————-

(J)

R2, Sections 5 and 6: It is not clear, how the polynomial of 4th order has been obtained and why this order has been chosen (a simple interpolation might be more straight-forward to implement).

——————

A 4th-order polynomial provided a smooth, accurate fit through points generated at discrete windspeeds.

——————

Proposed resolution:

Add text to this effect in Section 6.

————————————————————————-

The "less important issues" identified by Reviewer 2 will be incorporated.

---

## Author Response (AR1)

I have revised the manuscript along the point-by-point proposals in my initial response to reviewer comments. Most significantly, an introductory paragraph has been added which clarifies the context and motivation of the manuscript, related to reference wind turbine controllers. The principal conclusion is now limited to a reference PI controller, as opposed to controllers of a more general class. I hope that this satisfies the principal objections of the reviewers related to the scope of the findings.